# Efficacy of Nrf2 activation in a proteinuric Alport syndrome mouse model

Shota Kaseda[1,2] , Jun Horizono[1], Yuya Sannomiya[1], Jun Kuwazuru[1], Mary Ann Suico[1,3] , Ryoichi Sato[1], Hirohiko Fukiya[4], Hidetoshi Sunamoto[4] , Sayaka Ogi[4], Takashi Matsushita[4] , Yuimi Koyama[1], Aimi Owaki[1], Haruki Tsuhako[1], Masahiro Shiraga[1], Hiroshi Watanabe[5], Takehiro Nakano[6], Bernard Davenport[2] , Kandai Nozu[7], Masayuki Yamamoto[8], Tsuyoshi Shuto[1,3], Yasunori Tokunaga[4], Rachel Lennon[2] , Kazuhiro Onuma[4] , Hirofumi Kai[1,3]

Activation of nuclear factor erythroid 2–related factor 2 (Nrf2) has shown protective effects in experimental models of acute kidney injury and nonproteinuric chronic kidney disease. However, the efficacy of Nrf2 activation for proteinuric chronic kidney disease with glomerular injury is controversial, as a transient increase in proteinuria is observed. Here, we identified a potent Nrf2 activator UD-051, which inhibits the interaction between Kelch-like ECH-associated protein 1 (Keap1) and Nrf2. UD-051 significantly ameliorated the progressive phenotype of Alport syndrome mouse model in an Nrf2-dependent manner, accompanied by increased proteinuria. Mild Nrf2 activation by genetic *Keap1* knockdown or pharmacological Keap1 inhibition with CDDO-imidazolide did not attenuate Alport kidney disease, suggesting that strong Nrf2 activation is essential for clear therapeutic efficacy. In-depth analysis revealed that UD-051 suppressed tubular injury, including oxidative stress, inflammation, and dys-regulated metabolism. UD-051 with losartan, a renin–angiotensin system inhibitor that targets glomerular dysfunction, vastly ameliorated Alport kidney disease. Our study provides a comprehensive insight into the efficacy of Nrf2 activation in Alport syndrome and provides a rationale for adding a Keap1-Nrf2 interaction inhibitor to a renin–angiotensin system inhibitor.

## Introduction

The Kelch-like ECH-associated protein 1 (Keap1) and the nuclear factor erythroid 2–related factor 2 (Nrf2) system is pivotal in the protective response to cellular stress (Yagishita et al, 2014). Under normal conditions, Nrf2 is ubiquitinated by CUL3-Keap1 ubiquitin E3 ligase complex and degraded through the proteasomal pathway (Itoh et al, 1999). Upon exposure to reactive oxygen stress or electrophiles, reactive cysteine residues of Keap1 are modified and Nrf2 ubiquitination is halted (Kobayashi et al, 2004). This stabilizes and translocates Nrf2 to the nucleus, inducing the expression of cytoprotective detoxification and antioxidant enzymes (Yamamoto et al, 2018). The broad spectrum of physiological roles regulated by Nrf2 suggests that the Keap1-Nrf2 system is an important therapeutic target for intractable diseases (Lin et al, 2023).

Accumulating evidence suggests that Nrf2 activation has protective effects in numerous experimental models of acute kidney injury and nonproteinuric chronic kidney disease (CKD) with mainly tubular injury (Bondi et al, 2024). However, the efficacy of Nrf2 activators for proteinuric CKD with glomerular injury is controversial. Many animal studies indicated ameliorative effects (Jiang et al, 2014; Wu et al, 2014; Nagasu et al, 2019; Lu et al, 2020; Xu et al, 2023), whereas several suggested exacerbations (Zoja et al, 2013; Vaziri et al, 2015; Rush et al, 2021) manifested by increased proteinuria and kidney tissue injury. Although bardoxolone methyl had been tested for CKD patients, including Alport syndrome (Chertow et al, 2021), the Food and Drug Administration declined a new drug application because of concerns about its long-term efficacy and safety profile. However, no reports evaluated bardoxolone methyl or its derivatives in an Alport syndrome mouse model, and the detailed effects of Nrf2 activation on the Alport kidney have not been clarified.

We recently demonstrated that the Keap1-Nrf2 protein–protein interaction (PPI) inhibitor UBE-1099 transiently increased proteinuria, but attenuated kidney disease progression in the Alport syndrome mouse model (B6.Cg-*Col4a5*[tm1Yseg]/J) (Rheault et al, 2004; Kaseda et al, 2021). Distinct from covalent Nrf2 activators such as bardoxolone methyl, which irreversibly bind to the cysteine residue of Keap1, noncovalent Nrf2 activators reversibly inhibit Keap1-Nrf2

[1]Department of Molecular Medicine, Graduate School of Pharmaceutical Sciences, Kumamoto University, Kumamoto, Japan    [2]Wellcome Centre for Cell-Matrix Research, University of Manchester, Manchester, UK    [3]Global Center for Natural Resources Sciences, Faculty of Life Sciences, Kumamoto University, Kumamoto, Japan    [4]Pharmaceutical Research Laboratory, UBE Corporation, Yamaguchi, Japan    [5]Department of Clinical Pharmacy and Therapeutics, Graduate School of Pharmaceutical Sciences, Kumamoto University, Kumamoto, Japan    [6]Department of Biopharmaceutics, Graduate School of Pharmaceutical Sciences, Kumamoto University, Kumamoto, Japan    [7]Department of Pediatrics, Kobe University Graduate School of Medicine, Hyogo, Japan    [8]Tohoku Medical Megabank Organization, Tohoku University, Sendai, Japan

Correspondence: shota.kaseda@manchester.ac.uk; kazuhiro.onuma@ube.com; hirokai@gpo.kumamoto-u.ac.jp

PPI. Noncovalent Nrf2 activators present an innovative strategy that can enhance the selectivity of the agents and reduce the risk of side effects (Pallesen et al, 2018). However, the effect of UBE-1099 on survival in Alport mice is inferior to that of losartan (Omachi et al, 2021), an angiotensin II receptor blocker (ARB) used clinically for proteinuric CKD including Alport syndrome, suggesting that its effectiveness on the onset of end-stage renal disease in CKD patients is insufficient. Moreover, our previous pharmacological study was not able to clarify the direct relationship among Nrf2 activation, proteinuria, and kidney disease progression.

In this study, we developed a more potent Keap1-Nrf2 PPI inhibitor, UD-051, and revealed that UD-051 ameliorated the progressive phenotype in Alport mice with increased proteinuria, but not in *Nrf2* knockout (KO)-Alport mice. Importantly, the therapeutic efficacy of UD-051 was dose-dependent, and its maximum effectiveness exceeded that of losartan. In contrast, mild Nrf2 activation by genetic *Keap1* knockdown (KD) or rodent tolerable bardoxolone methyl analog, CDDO-imidazolide (CDDO-Im), did not attenuate kidney disease progression in Alport mice, suggesting that strong Nrf2 activation is essential for clear therapeutic efficacy. Moreover, the therapeutic efficacy of UD-051 was significantly enhanced when combined with losartan. Together, our study provides a comprehensive insight into the direct relationship among Nrf2 activation, proteinuria, and kidney disease progression in Alport syndrome, and indicates better efficacy of adding a Keap1-Nrf2 PPI inhibitor to ARB.

# Results

## Nrf2 activation by Keap1 knockdown or CDDO-Im did not attenuate the progressive phenotype in the Alport mice

To assess the effect of genetic Nrf2 activation in proteinuric CKD with glomerular injury, we crossed Alport mice with *Keap1* KD mice (Okawa et al, 2006; Taguchi et al, 2010) and examined kidney function and pathology in homozygous *Keap1* KD-Alport (Fig 1A). *Keap1* KD mice have low Keap1 mRNA expression and a constitutively induced expression of Nrf2 target molecule NAD(P)H: quinone oxidoreductase 1 (NQO1) in kidney tissue (Fig 1B and C). *Keap1* KD and *Keap1* KD-Alport mice had low weight gain compared with WT and Alport mice, respectively. Urine volume decreased at the early stage and increased at the late stage in *Keap1* KD-Alport mice, but no noticeable toxicity was suspected (Fig S1A and B). *Keap1* KD slightly increased proteinuria, but did not affect GFR and plasma creatinine in Alport mice (Fig 1D–F), suggesting that *Keap1* KD did not ameliorate kidney dysfunction. Next, we analyzed the histopathology by PAS staining for glomerular injury, immunostaining of F4/80-positive macrophages for inflammation, and Masson's trichrome staining for fibrosis. Alport mice exhibited typical glomerulosclerosis, macrophage infiltration, and interstitial fibrosis (Fig 1G–J). *Keap1* KD did not attenuate these pathologies or the dysregulation of mRNA expression related to inflammation, fibrosis, and kidney injury (Fig S1C–I). Furthermore, *Keap1* KD did not extend the lifespan of Alport mice (Fig 1K). We hypothesized that the low effectiveness of *Keap1* KD is due to the genetic constitutive Nrf2

activation or its insufficient strength. Therefore, we treated Alport mice with 3 or 10 mg/kg of CDDO-Im to induce pharmacologic Nrf2 activation (Fig S2A). The intensity of Nrf2 activity induced by CDDO-Im at 3 mg/kg, estimated from the maximal induction of Nqo1 mRNA expression, is comparable to that in *Keap1* KD mice (Kaseda et al, 2021). CDDO-Im did not affect body weight and urine volume in Alport mice, but slightly increased proteinuria at 3 mg/kg and did not attenuate kidney dysfunction (Fig S2B–F). Although CDDO-Im did not ameliorate the glomerulosclerosis and fibrosis in Alport mice, it significantly suppressed macrophage infiltration (Fig S2G–J). Together, CDDO-Im suppressed kidney inflammation, especially inflammatory cytokines in a dose-dependent manner (Fig S2K–P), but not kidney disease progression in Alport mice. These results suggest that the stronger Nrf2 activity is more essential than its constant activation for suppressing the progressive phenotype in Alport mice.

### Discovery of a Keap1-Nrf2 protein–protein interaction inhibitor UD-051

Aiming to find a more potent Nrf2 activator, we used UBE-1099 as a lead compound and conducted a fluorescence polarization–based screening to optimize its structure (Kaseda et al, 2021). We identified UD-051 that inhibited the Keap1-Nrf2 PPI and activated NQO1 in Hepa1c1c7 cells (Fig S3A–C). Knockdown of *Nrf2* abolished the UD-051–induced mRNA expression of Nqo1 (Fig S3D and E). UD-051 has high oral absorption, especially in C57BL/6J mice and cynomolgus monkeys, with bioavailability of over 90% (Fig S3F–J). UD-051 increased the Nqo1 mRNA in murine kidney dose-dependently up to 3 mg/kg (Fig S3K). NQO1 protein in mouse and rat kidney tissue was also increased (Fig S3L and M). The maximum intensity of Nrf2 activity induced by UD-051 at 0.3 mg/kg was comparable to the CDDO-Im at 10 mg/kg, and UD-051 at 1 mg/kg was comparable to the UBE-1099 at 30 mg/kg (Kaseda et al, 2021). Together, these data indicate that we successfully identified a novel Keap1-Nrf2 PPI inhibitor with high activity and excellent oral absorption.

### UD-051 suppressed kidney injury in Alport mice

To assess the effect of UD-051 in vivo, we treated Alport mice with 0.3, 1, or 3 mg/kg of UD-051 or vehicle and examined the kidney function and pathology (Fig 2A). Slight suppression of weight gain and transient increase in urine volume were observed in UD-051–treated Alport mice, but no noticeable toxicity was suspected (Fig S4A and B). UD-051 prevented the decline of GFR, and the increase in plasma creatinine, blood urea nitrogen (BUN), and indoxyl sulfate in Alport mice (Fig 2B–E), suggesting that UD-051 dose-dependently ameliorated kidney dysfunction. PAS staining revealed that >50% of glomeruli showed severe glomerulosclerosis (score 4) in vehicle-treated Alport mice that was reduced to mild glomerulosclerosis (score 2) with thickening of Bowman's capsule by treatment with UD-051 (Fig 2F and G). UD-051 markedly attenuated the macrophage infiltration and fibrosis area (Fig 2F, H, and I). UD-051 suppressed the increase in α-smooth muscle actin (α-SMA)–positive glomerular crescents and myofibroblast area, and the kidney injury molecule (KIM)-1–positive tubules. UD-051 also suppressed the decrease in lotus tetragonolobus lectin (LTL)–

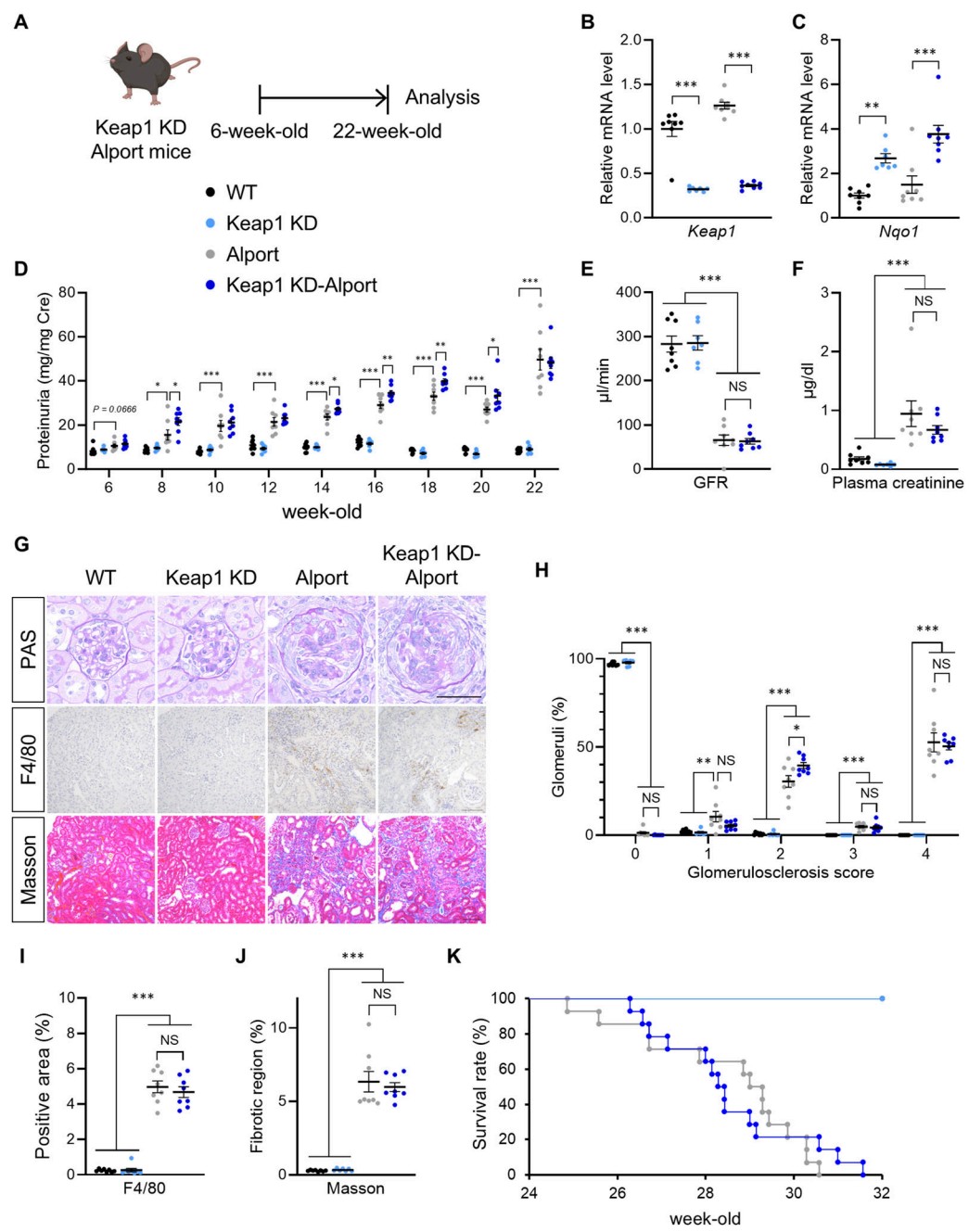

**Figure 1.** *Keap1* **knockdown neither attenuated the disease progression nor prolonged the lifespan of Alport mice.**
**(A)** Experimental design for *Keap1* KD-Alport mice. **(B, C)** Relative expression level of the indicated mRNA in kidneys from WT, *Keap1* KD, Alport, and *Keap1* KD-Alport mice. **(D)** Urinary protein concentration normalized to creatinine concentration in indicated time points of WT, *Keap1* KD, Alport, and *Keap1* KD-Alport mice. **(E, F)** GFR and plasma creatinine in 22-wk-old WT, *Keap1* KD, Alport, and *Keap1* KD-Alport mice. **(G)** Representative images of PAS staining, immunohistochemistry of F4/80, and Masson's trichrome staining of kidney sections from WT, *Keap1* KD, Alport, and *Keap1* KD-Alport mice. Scale bars: 50 μm (for PAS staining) and 100 μm (for other staining). **(H)** Glomerulosclerosis scores based on the PAS-stained sections. **(I, J)** Percentage of F4/80-positive area and fibrotic region in kidney sections. Data are presented as the mean ± SEM (n = 7–8 per group). *P*-values were assessed by Tukey's multiple comparisons test. \*P < 0.05, \*\*P < 0.01, \*\*\*P < 0.001. NS, not significant. **(K)** Kaplan–Meier survival curves. Results were derived from WT mice (n = 14), *Keap1* KD mice (n = 14), Alport mice (n = 14), and *Keap1* KD-Alport mice (n = 14).

positive proximal tubules in Alport mice (Fig 2F and J–L). Consistently, UD-51 down-regulated the genes related to inflammation, fibrosis, and kidney injury (Fig S5A–J). Collectively, these results suggest that UD-051 dose-dependently suppressed kidney injury in Alport mice.

## UD-051 suppressed urinary biomarkers in Alport mice accompanied by proteinuria

Urinary cystatin C, $\beta_2$-microglobulin, neutrophil gelatinase-associated lipocalin (NGAL), KIM-1, clusterin, and trefoil factor 3

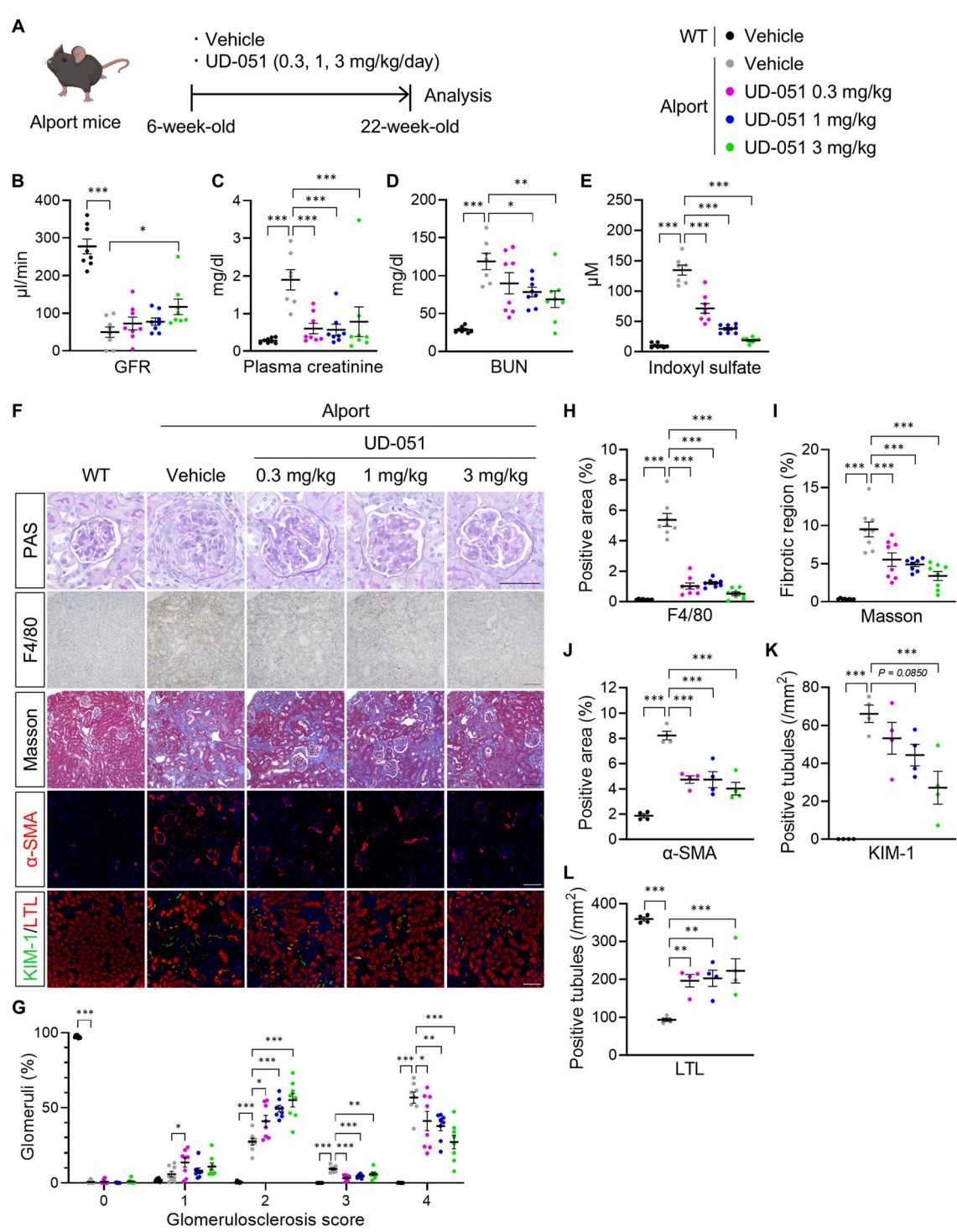

**Figure 2. Protective effects of UD-051 against kidney injury in Alport mice.**
**(A)** Experimental design for administration of UD-051 in Alport mice. **(B, C, D, E)** GFR, plasma creatinine, BUN, and indoxyl sulfate in 22-wk-old WT and Alport mice.
**(F)** Representative images of PAS staining, immunohistochemistry of F4/80, Masson's trichrome staining, and immunofluorescence of α-SMA, KIM-1, and lotus tetragonolobus lectin of kidney sections in WT and Alport mice. Scale bars: 50 μm (for PAS staining) and 100 μm (for other staining). **(G)** Glomerulosclerosis scores based on the PAS-stained sections. **(H, I)** Percentage of F4/80-positive area and fibrotic region in kidney sections. Data are presented as the mean ± SEM (n = 7–8 per group). P-values were assessed by Dunnett's multiple comparisons test. *P < 0.05, **P < 0.01, ***P < 0.001. **(J)** Percentage of α-SMA–positive area in kidney sections. **(K, L)** Number of KIM-1– and lotus tetragonolobus lectin–positive tubules in a 1-mm² area. Data are presented as the mean ± SEM (n = 4 per group). P-values were assessed by Dunnett's multiple comparisons test. **P < 0.01, ***P < 0.001.

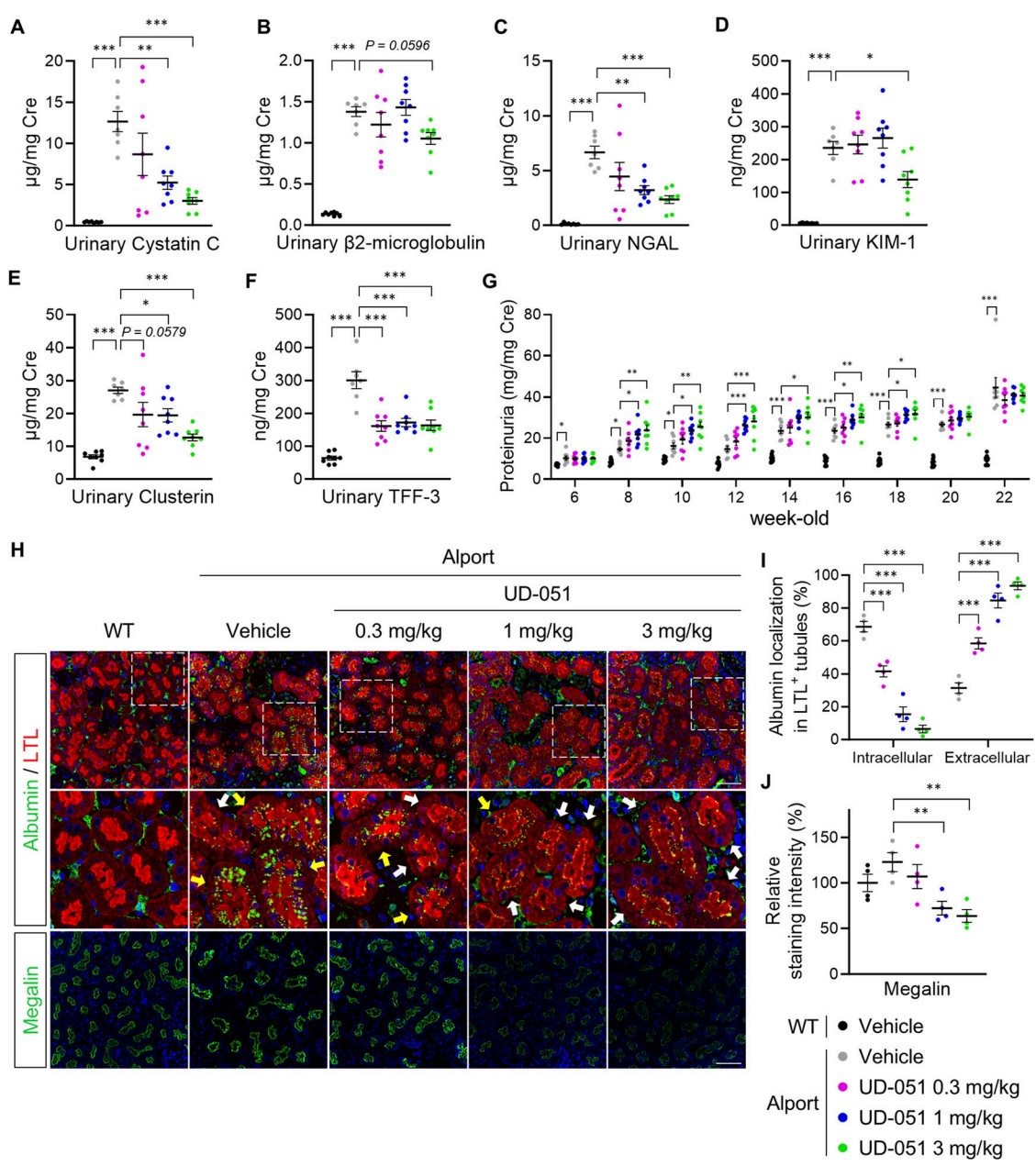

**Figure 3. UD-051 suppressed urinary biomarkers in Alport mice accompanied by proteinuria.**
**(A, B, C, D, E, F)** Urinary cystatin C, $\beta_2$-microglobulin, NGAL, KIM-1, clusterin, and TFF3 concentrations normalized to creatinine concentration in 22-wk-old WT and Alport mice. **(G)** Urinary protein normalized to creatinine concentration at the indicated time points of WT and Alport mice. Data are presented as the mean ± SEM (n = 7–8 per group). *P*-values were assessed by Dunnett's multiple comparisons test. *$P < 0.05$, **$P < 0.01$, ***$P < 0.001$ versus Alport vehicle. **(H)** Representative images of immunofluorescence for albumin, lotus tetragonolobus lectin, and megalin in 22-wk-old WT and Alport mice. Yellow arrows indicate proximal tubules with intracellular localization of albumin (weak red area = cell body), and white arrows indicate proximal tubules with extracellular localization of albumin (strong red area = brush border). Scale bars: 100 $\mu m$. **(I)** Percentage of proximal tubules with intracellular or extracellular (intraluminal) albumin accumulation. **(J)** Relative staining intensity of megalin in kidney sections. Data are presented as the mean ± SEM (n = 4 per group). *P*-values were assessed by Dunnett's multiple comparisons test. **$P < 0.01$, ***$P < 0.001$.

(TFF3) indicate tubular injury (Zhang & Parikh, 2019), and proteinuria and albuminuria indicate glomerular damage (Moeller & Chia-Gil, 2020). We measured these parameters to check the efficacy of UD-051 on kidney dysfunction in Alport mice. Consistent with the protective effect of UD-051 against tubular injury, UD-051 markedly reduced the levels of urinary cystatin C, NGAL, clusterin, and TFF3 and slightly suppressed $\beta_2$-microglobulin and KIM-1 at

22 wk of age (Fig 3A–F). Paradoxically, similar to the clinical trial of bardoxolone methyl (Rossing et al, 2019), UD-051 increased proteinuria in Alport mice from 8 to 20 wk old (Fig 3G). It was reported that bardoxolone methyl decreases the expression of megalin (Reisman et al, 2012), an endocytic receptor involved in the reabsorption of albumin in the proximal tubule, suggesting that Nrf2 activation promotes albumin excretion in the urine. We examined

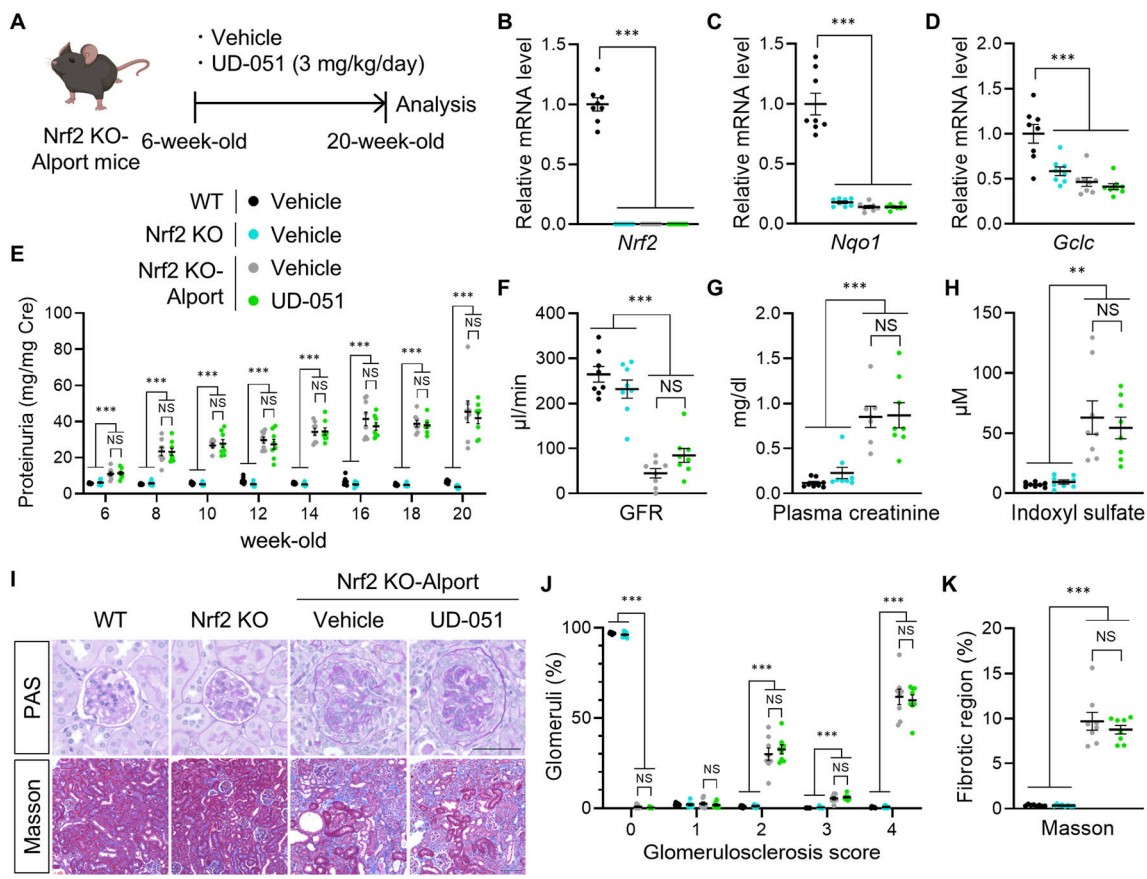

**Figure 4. Effects of UD-051 were not observed in *Nrf2* knockout-Alport mice.**
**(A)** Experimental design for administration of UD-051 in *Nrf2* KO-Alport mice. **(B, C, D)** Relative expression level of the indicated mRNA in the kidneys of WT, *Nrf2* KO, and *Nrf2* KO-Alport mice. **(E)** Urinary protein concentration normalized to creatinine concentration at the indicated time points in WT, *Nrf2* KO, and *Nrf2* KO-Alport mice. **(F, G, H)** GFR, plasma creatinine, and indoxyl sulfate in 20-wk-old WT, *Nrf2* KO, and *Nrf2* KO-Alport mice. **(I)** Representative images of PAS staining and Masson's trichrome staining of kidney sections in WT, *Nrf2* KO, and *Nrf2* KO-Alport mice. Scale bars: 50 μm (for PAS staining) and 100 μm (for Masson's trichrome staining). **(J)** Glomerulosclerosis scores based on the PAS-stained sections. **(K)** Percentage of fibrotic region in Masson's trichrome staining sections. Data are presented as the mean ± SEM (n = 8 per group). *P*-values were assessed by Tukey's multiple comparisons. *P < 0.05, **P < 0.01, ***P < 0.001.

the proximal tubule uptake of albumin and the megalin expression level by immunofluorescence. Albumin was localized intracellularly in 75% of proximal tubules (yellow arrow) and extracellularly in 25% of proximal tubules (white arrow—intraluminal) in vehicle-treated Alport mice, but these ratios were reversed in a UD-051 dose-dependent manner (Fig 3H and I). Moreover, UD-051 decreased the expression of megalin, but not cubilin and CD36 (Figs 3H and J and S6A–C) (Nielsen et al, 2016). The decrease in the megalin level explains the increased proteinuria observed in UD-051–treated Alport mice. Contrary to the conventional concept for proteinuric CKD treatment, UD-051 attenuated the progression of kidney dysfunction with pharmacological proteinuria.

### The effects of UD-051 were not observed in *Nrf2* knockout-Alport mice

To determine whether Nrf2 mediates the effects of UD-051, we generated *Nrf2* KO-Alport mice and treated them with UD-051 (Fig 4A). UD-051 did not induce the mRNA expression of Nqo1 and Gclc in *Nrf2* KO-Alport mice (Fig 4B–D). UD-051 neither increased the

proteinuria nor ameliorated the pathologies in *Nrf2* KO-Alport mice (Figs 4E–K and S7A–E). The UD-051–induced suppression of weight gain and transient increase in urine volume in Alport mice were also abolished in *Nrf2* KO-Alport mice (Fig S7F and G). Conversely, a slight, but nonstatistical, increase in body weight was observed in *Nrf2* KO mice compared with WT mice. Considering these results and low weight gain in *Keap1* KD mice (Fig S1A), the UD-051–induced suppression of weight gain is not due to an off-target side effect of the compound but is due to the on-target effect of Nrf2. Together, these results demonstrate that Nrf2 is essential for the therapeutic effect and increase in proteinuria of UD-051 in Alport mice.

### Transcriptome analysis revealed the molecular effects of UD-051 in Alport kidneys

Alport mice develop spontaneous glomerulosclerosis and exhibit secondary tubular injury in kidney tissue (Kaseda et al, 2021; Omachi et al, 2021; Sannomiya et al, 2021). To explore the molecular mechanism of UD-051 on glomerulosclerosis and tubular injury, we treated Alport mice with UD-051 and performed RNA sequencing of

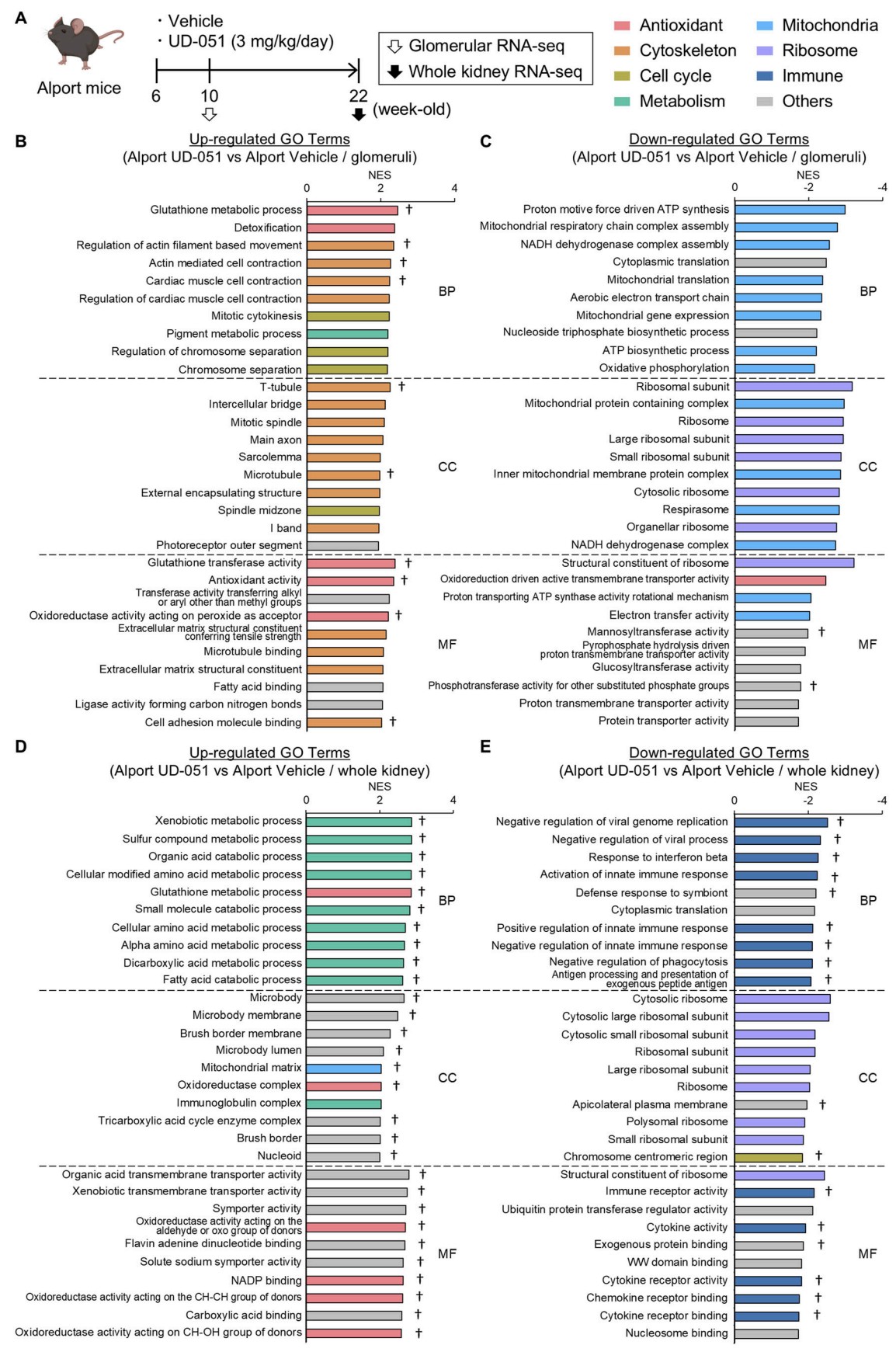

glomeruli at 10 wk of age, and whole kidney at 22 wk of age (Fig 5A). Gene set enrichment analysis (GSEA) identified an increase in antioxidant-related gene ontology (GO) and a decrease in ribosome-related GO as common targets of UD-051 in the glomeruli and whole kidney (Fig 5B–E). Nrf2 major target molecules, NQO1 and glutamate–cysteine ligase catalytic subunit (GCLC), were increased in the kidney tissue of UD-051–treated Alport mice (Fig S8A–F). Although the role of Nrf2 in anti-oxidative response is well studied, the link between Nrf2 and ribosomes is less well known. It is speculated that ribosome biogenesis (Hirai et al, 2022) is suppressed by Nrf2 to control energy homeostasis. GSEA identified Nrf2 downstream pathways such as cytoskeleton (Ko et al, 2021), cell cycle (Reddy et al, 2008; Homma et al, 2009), and mitochondria (Dinkova-Kostova & Abramov, 2015) in glomeruli (Fig 5B and C). In the whole kidney, GSEA identified metabolism (Uruno et al, 2013) and immunity (Saha et al, 2020) as the targets of UD-051(Fig 5D and E). Many of the GOs altered by UD-051 in the glomerulus were not dysregulated in the Alport vehicle. In contrast, many of the GO altered by UD-051 in the whole kidney were inversely correlated with the GO (Dagger) altered in Alport vehicle, such as the metabolic processes and immune response pathways (Figs 5D and E, S9A–D, and S10A–D). Overall, these results suggest that UD-051 ameliorated glomerulosclerosis through activation of Nrf2 and its downstream pathways, not through the improvement of signaling dysregulation in Alport glomeruli, and suppressed subsequent tubular injury.

### Combination intervention with UD-051 and losartan had additive therapeutic effects in Alport mice

Considering the different therapeutic mechanisms of UD-051 and ARB on kidney disease, we investigated whether UD-051 has an additive effect on losartan (Gross et al, 2012). Alport mice were treated with 0.3, 1, or 3 mg/kg of UD-051, losartan (Omachi et al, 2021), or both, and kidney function and survival span were examined (Fig 6A). UD-051, and the combination of losartan and UD-051 slightly suppressed weight gain in Alport mice, but no noticeable toxicity was suspected (Fig S11A). Combination therapy of losartan and 3 mg/kg of UD-051 significantly suppressed the increase in urine volume of Alport mice (Fig S11B). Compared with vehicle-treated Alport mice, UD-051 extended the median lifespan by 3.9%, 9.4%, and 22.7% at 0.3, 1, and 3 mg/kg, respectively, whereas losartan increased it by 8.4% (Fig 6B and C). Notably, UD-051 combined with losartan prominently prolonged the median lifespan of Alport mice by 15.8%, 33%, and 66% at 0.3, 1, and 3 mg/kg, respectively, showing superiority over monotherapy. Losartan showed an antiproteinuric effect in Alport mice, and losartan also suppressed the UD-051–induced increase in proteinuria (Fig 6D). Moreover, UD-051 was more effective in suppressing plasma creatinine in Alport mice, whereas losartan was more effective in suppressing urinary biomarkers (Figs 6E and S11C–F). The combination therapy further suppressed each parameter.

In a similar manner of treatment (Fig 6F), each monotherapy, slightly but not statistically, suppressed the decline of GFR in Alport mice (Fig 6G). Notably, combination therapy maintained GFR at the same level as WT mice. Glomerulosclerosis and fibrosis were ameliorated by UD-051 and losartan and further suppressed by combination therapy (Fig 6H–J). Transmission electron microscopy (TEM) revealed substantial glomerular basement membrane (GBM) thickening with podocyte foot process (FP) effacement in vehicle-treated Alport mice (Fig 6H, K–M) (Fukuda et al, 2016; Randles et al, 2016). UD-051, losartan, and combination therapy reduced the frequency of severe pathologies and suppressed the decline of podocyte FP density.

To check the efficacy of combination therapy in advanced CKD, we administered losartan alone or losartan and UD-051 to Alport mice at 12 wk of age when mice presented more severe pathology and proteinuria than at 6 wk, and examined kidney function at 24 wk (Fig S12A). Vehicle-treated Alport mice showed a GFR of 10.9% of WT mice, and losartan improved it by only 6.3%, whereas the combination therapy improved it by 44.6% (Fig S12B). Losartan only slightly decreased plasma creatinine and indoxyl sulfate, whereas the combination therapy greatly reduced these parameters (Fig S12C and D). Losartan significantly suppressed proteinuria at 18 wk of age, but did not prevent the subsequent increase in later stages (Fig S12E). In contrast, combination therapy significantly reduced the increase in proteinuria at 24 wk. These results revealed the strong additive effect of UD-051 to losartan therapy on the progressive phenotype in Alport mice.

### Oxidized albumin ratio inversely correlates highly with Nrf2 activity in Alport kidneys

Although our results showed that the intensity of Nrf2 activity in kidney tissue is essential to obtain appropriate drug efficacy, there is no minimally invasive method to assess it. Previously, we developed a quantitative measurement system for plasma oxidative stress using Cys34-cysteinylated albumin (oxidized albumin) (Nagumo et al, 2014; Watanabe et al, 2017), which is also clinically useful as a biomarker for the progression of kidney diseases (Imafuku et al, 2021). To investigate whether oxidized albumin reflects Nrf2 activity in kidney tissue, we administered UD-051 to Alport mice or *Nrf2* KO-Alport mice and measured plasma oxidized albumin ratio (OAR) and the mRNA expression of Nqo1 and Gclc in kidney tissue (Fig 7A). OAR was decreased in UD-051–treated Alport mice and increased in *Nrf2* KO-Alport mice compared with vehicle-treated Alport mice (Fig 7B). Importantly, OAR inversely correlated with Nqo1 and Gclc expression levels in Alport kidney tissue (Fig 7C and D). UD-051 dose-dependently decreased the OAR (Fig 7E and F), which showed an inverse correlation with the expression levels of Nqo1 and Gclc (Fig 7G and H). These results indicate that the OAR is a minimally invasive and highly sensitive biomarker that reflects Nrf2 activity in kidney tissue.

---

**Figure 5. Transcriptome analysis revealed the comprehensive effects of UD-051 in the whole kidney and glomeruli of the Alport mice.**
**(A)** Experimental design for glomerular and whole kidney RNA-seq in Alport mice. **(B, C, D, E)** Top 10 GO terms in Biological Process (BP), Cellular Component (CC), and Molecular Function (MF) analyzed by Gene Set Enrichment Analysis (v4.3.2). † = GO inversely correlated with changes in Alport vehicle versus WT (Fig S10).

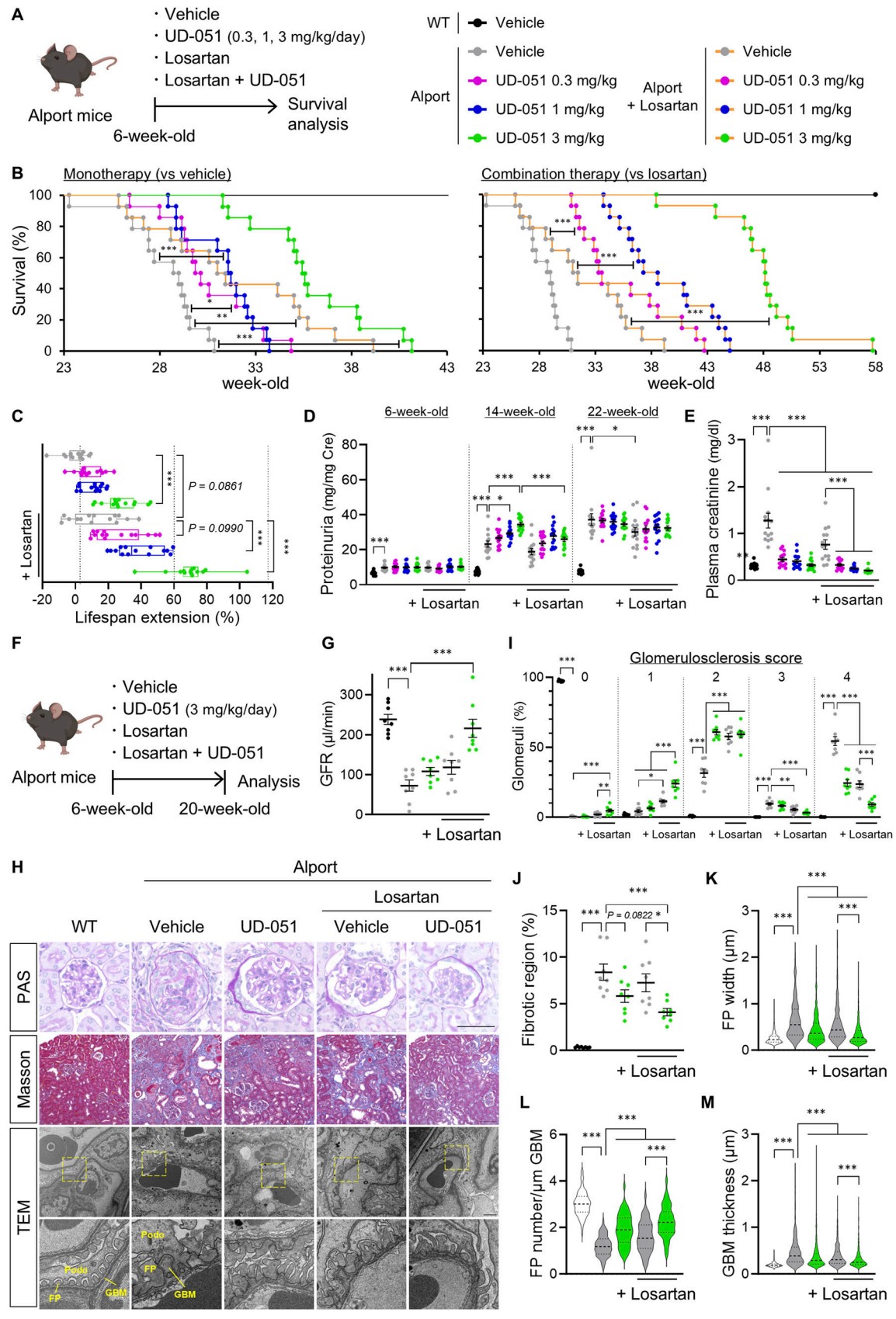

# Discussion

In this study, we revealed that mild Nrf2 activation by CDDO-Im and *Keap1*-KD did not suppress the progressive phenotype in Alport mice, whereas a potent Nrf2 activator UD-051 ameliorated it in an Nrf2-dependent manner. The therapeutic efficacy of UD-051 was significantly enhanced when combined with losartan. Notably, in contrast to the conventional therapeutic concept for proteinuric CKD, UD-051 ameliorated kidney dysfunction with pharmacological proteinuria.

In Alport syndrome, disruption of the GBM caused by variants in *COL4A3*, *COL4A4*, or *COL4A5* causes podocyte dysfunction and proteinuria, proliferation of Bowman's epithelial cells, and progressive glomerulosclerosis. UD-051 dose-dependently attenuated the glomerulosclerosis in Alport mice (score 2), but did not increase the proportion of glomeruli showing mild sclerosis (score 1) (Fig 2G). In contrast to a partial protective effect in the glomerulus, UD-051 significantly ameliorated tubular injury, inflammation, fibrosis, and abnormalities of gene expression in whole kidney tissue. Although UD-051 was inferior to losartan in suppressing the decline in GFR and progression of glomerulosclerosis, it inhibited the onset of subsequent fibrosis and end-stage renal failure more than losartan (Fig 6B, G–J). These results suggest that the main therapeutic target for UD-051 is the tubular injury common to most CKDs, rather than the glomerular filtration defects characteristic of some glomerular diseases, including Alport syndrome.

Although blockade of the renin–angiotensin system (RAS) is a current therapeutic approach for proteinuric CKD, including Alport syndrome, the mechanism of kidney disease progression is complex and cannot be completely ameliorated by RAS blockade only. We previously revealed that losartan suppressed podocyte abnormality in glomeruli of Alport mice, but not the dysregulation of genes related to metabolism, inflammation, and oxidative stress in the whole kidney (Omachi et al, 2021). Because losartan and UD-051 have different pharmacological features and targets, their combination could have additive effects on Alport kidney disease. Losartan was more effective against proteinuria and glomerular injury, whereas UD-051 was more effective against tubular injury. The effect of losartan and UD-051 on median survival in Alport mice was considered synergistic because median survival was greater than the sum of each monotherapy. Moreover, when administered at the middle stage of the disease (12-wk-old), UD-051 and losartan together but not losartan alone ameliorated kidney dysfunction in Alport mice, suggesting that UD-051 and losartan have different therapeutic targets and mechanisms, which enable comprehensive control of Alport kidney disease.

Filtered albumin in the glomerulus is reabsorbed into the proximal tubule, causing injury and contributing to kidney disease progression (Jarad et al, 2016). Therefore, the current therapeutic strategy for proteinuric CKD is to suppress proteinuria (Gross et al, 2012). In contrast, UD-051 increased proteinuria in an Nrf2-dependent manner and ameliorated kidney disease progression. Clinical trials of bardoxolone universally showed a transient increase of proteinuria in recipients, including Alport syndrome, which remains a concern with Nrf2 activation (Warady et al, 2022). In-depth analysis suggests that UD-051 inhibits the albumin uptake in the proximal tubule by reducing the expression of megalin. Changes in the urinary cystatin C and $\beta_2$-microglobulin in UD-051–treated Alport mice support this finding. Cystatin C and $\beta_2$-microglobulin freely pass through glomerular filtration and are reabsorbed by megalin in the proximal tubule. Previous report showed that in megalin KO mice, the urinary excretion of $\beta_2$-microglobulin was ≥10x higher than cystatin C (Zhao et al, 2022). Correlating with this result, UD-051 significantly reduced urinary cystatin C in Alport mice, but only slightly reduced urinary $\beta_2$-microglobulin. Although further investigation is needed to clarify the mechanism, based on these reports and the current data, we hypothesize that the increase in proteinuria induced by UD-051 is nonpathologic, Nrf2-dependent, and independent of the progression of kidney disease.

Although many reports showed protective effects of Nrf2 activators on proteinuric CKD with glomerular injury (Jiang et al, 2014; Lu et al, 2020; Kaseda et al, 2021; Xu et al, 2023), other studies reported exacerbating effects (Vaziri et al, 2015; Rush et al, 2021). The reason for the divergent results is unclear. Multiple-target effects and mechanisms may be possible. Based on our observations, we hypothesize that the dosing for Nrf2 activation in kidney disease may be important. Low doses of bardoxolone methyl analog, RTA-dh404, attenuated glomerulosclerosis in 5/6 nephrectomy models, whereas high doses caused a worsening (Vaziri et al, 2015). Another bardoxolone methyl analog, RTA402, caused death in the aldosterone-induced hypertension mouse model when administered at high concentrations immediately after the start of the study (Hisamichi et al, 2017). However, RTA402 was well tolerated and showed protective effects when its concentration was gradually increased. We confirmed that UD-051–induced Nrf2 activity increased rapidly in the rat kidney tissue compared with mice, making it difficult to control Nrf2 activity (Fig S3L and M). This may be why exacerbation of Nrf2 activators is often reported in rat kidney disease models (Zoja et al, 2013; Vaziri et al, 2015). Moreover, because kidney failure progresses gradually over 6 mo in the Alport mice, it may have been possible to ensure a wide margin of safety for Nrf2 activity. Thus, strict control of Nrf2 activity in kidney tissue is

**Figure 6. Combination therapy of UD-051 and losartan attenuated disease progression and prolonged the lifespan of Alport mice.**
**(A)** Experimental design for administration of UD-051 or losartan alone, or in combination in Alport mice. **(B)** Kaplan–Meier survival curves of WT and Alport mice. *P*-values were assessed by the log-rank (Mantel–Cox) test. *$P < 0.05$, **$P < 0.01$, ***$P < 0.001$. **(C)** Percentage of lifespan extension (over vehicle-treated Alport mice) of WT and Alport mice. Data are shown as boxes and whiskers. **(D)** Urinary protein concentration normalized to creatinine concentration at the indicated time points in WT and Alport mice. **(E)** Plasma creatinine in 22-wk-old WT and Alport mice. Data are presented as the mean ± SEM (n = 14 per group). *P*-values were assessed by Tukey's multiple comparisons. *$P < 0.05$, **$P < 0.01$, ***$P < 0.001$. **(F)** Experimental design for administration of UD-051 or losartan alone, or in combination, in Alport mice. **(G)** GFR in 20-wk-old WT and Alport mice. **(H)** Representative images of PAS staining, Masson's trichrome staining, and TEM of kidney sections in WT and Alport mice. Scale bars: 50 $\mu$m (for PAS staining), 100 $\mu$m (for Masson's trichrome staining), and 2 $\mu$m (for TEM). **(I)** Glomerulosclerosis scores based on the PAS-stained sections. **(J)** Percentage of fibrotic region in Masson's trichrome staining sections. Data are presented as the mean ± SEM (n = 8 per group). **(G, H, I, J, K, L, M)** Colors represent WT (black), Alport vehicle (gray), and Alport UD-051 (green). *P*-values were assessed by Tukey's multiple comparisons. *$P < 0.05$, **$P < 0.01$, ***$P < 0.001$. **(K, L, M)** FP width, FP number, and GBM thickness based on TEM. Data are presented as violin plots (n = 3 per group). *P*-values were assessed by Tukey's multiple comparisons. *$P < 0.05$, ***$P < 0.001$.

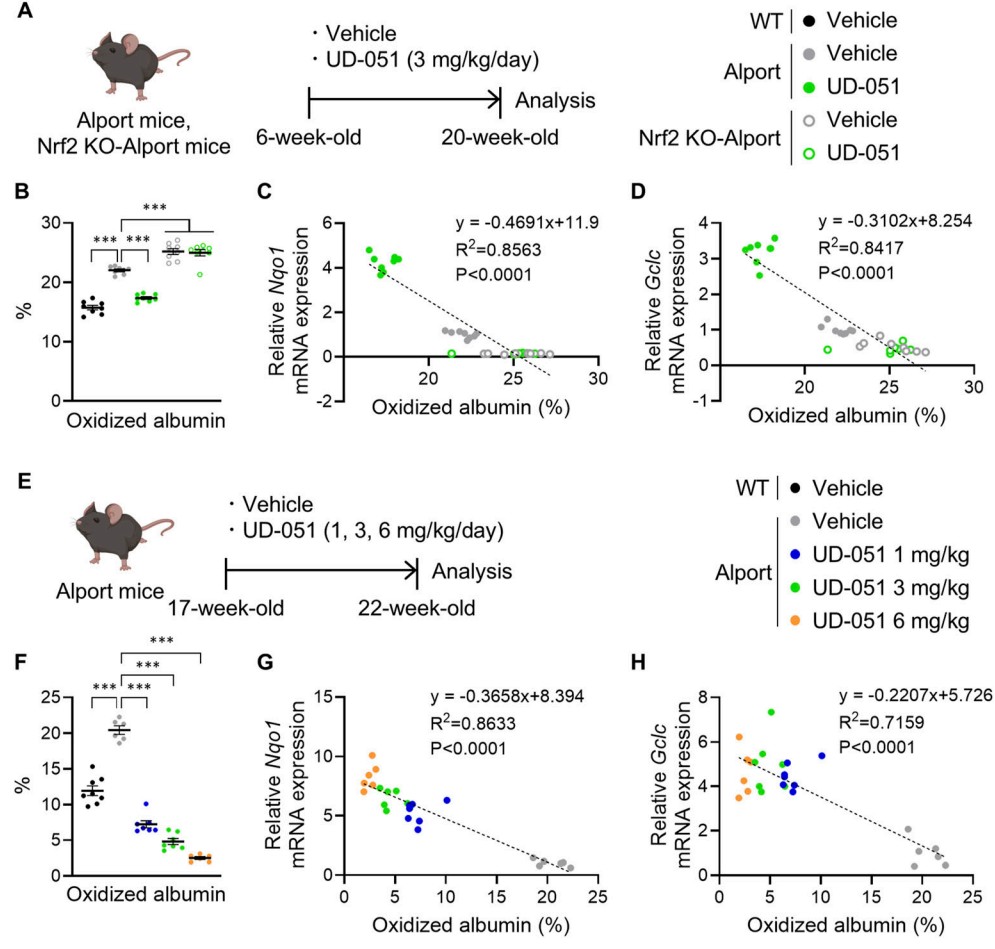

**Figure 7. Correlation analysis between plasma oxidized albumin ratio and Nrf2 activity in the Alport kidney tissue.**
**(A)** Experimental design for administration of UD-051 in Alport mice or *Nrf2* KO-Alport mice. **(B)** Oxidized albumin ratio (OAR) in 20-wk-old WT, Alport mice, and *Nrf2* KO-Alport mice. Data are presented as the mean ± SEM (n = 8 per group). *P*-values were assessed by Tukey's multiple comparisons. ***$P$ < 0.001. **(C, D)** Correlation between OAR in plasma and mRNA levels of Nqo1 and Gclc in the kidney. *P*-values and the coefficient of determination ($R^2$) were assessed by the Pearson correlation coefficient. **(E)** Experimental design for administration of UD-051 in Alport mice. **(F)** OAR in 22-wk-old WT and Alport mice. Data are presented as the mean ± SEM (n = 6–7 per group). *P*-values were assessed by Tukey's multiple comparisons test. ***$P$ < 0.001. **(G, H)** Correlation between OAR in plasma and mRNA levels of Nqo1 and Gclc in the kidney. *P*-values and the coefficient of determination ($R^2$) were assessed by the Pearson correlation coefficient.

important to obtain protective effects. We previously showed that the OAR is a marker for predicting the progression of CKD (Imafuku et al, 2021), and it is expected that it will become a promising minimally invasive biomarker in the clinical development of Nrf2 activators and will enable dose titration.

Importantly, UD-051 shows high oral absorption in multiple animal species, especially in cynomolgus monkeys, with bioavailability of over 90%, and is expected to have similarly high activity in humans. Although further studies, such as the relationship between the effect of Nrf2 activation and its threshold in each kidney disease model, are required, future clinical application of Keap1-Nrf2 PPI inhibitor to CKD is expected.

Overall, these results provide a comprehensive insight into the ameliorative effect of Nrf2 in Alport syndrome and may indicate better efficacy of adding a Keap1-Nrf2 PPI inhibitor to the RAS inhibitor.

# Materials and Methods

### Animal experiments

Alport mice (B6.Cg-*Col4a5*$^{tm1Yseg}$/J, strain #006183) (Rheault et al, 2004), *Keap1* knockdown mice (B6.129P2-*Keap1*$^{tm2Mym}$, BRC #RBRC09595) (Okawa et al, 2006; Taguchi et al, 2010), *Nrf2* knockout mice (B6.129X1-*Nfe2l2*$^{tm1Ywk}$/J, strain #017009) (Chan et al, 1996), and age-matched C57BL/6 mice (Charles River Laboratories) were used. In all experiments, male mice were used to avoid sex differences.

To minimize the individual and experimental variance, all Alport mice within the same study were generated at once from frozen embryos, stratified by proteinuria score (7–16 mg/mg Cre for 6-wk-old and 11.5–32 mg/mg Cre for 12-wk-old) and body weight, and randomly assigned to experimental groups using stratified randomization (Exsus ver 10.0). 0.5 wt/vol% methylcellulose (vehicle) (133-17815, Fujifilm), UD-051 (synthesized by Pharmaceutical Research Laboratory, UBE Corporation), or CDDO-Im (HY-15725; MedChemExpress) was orally administered at the indicated dose. Losartan (L0232; TGI) was administered via drinking water at 250 μg/ml to 6- to 11-wk-old Alport mice, and 125 μg/ml to 12-wk-old or older Alport mice. Alport mice exhibit increased water intake as the disease progresses. Therefore, we reduced the concentration of losartan accordingly (Yokota et al, 2018; Kaseda et al, 2021; Omachi et al, 2021; Sannomiya et al, 2021). The water containing losartan was replenished twice per week. Sample sizes, the number of replicates, and the grouping threshold were determined based on pilot experiments and our previous results (Yokota et al, 2018; Kaseda et al, 2021; Omachi et al, 2021; Sannomiya et al, 2021). Although each

animal experiment was performed only once, the results were substantiated by repetition under various conditions. Investigators were not blinded, but pathological analysis was performed in a blind fashion. No animals were excluded from the study, but some samples subsequently failed quality control in the assay (e.g., for RNA quality/abundance, urine, and blood sample volume).

For measurement of pharmacokinetics, the indicated concentration of UD-051 was administered intravenously or orally to 8-wk-old male C57BL/6J mice, 6-wk-old female BALB/c mice, 7-wk-old male CD (SD) rats, and 9-yr-old male cynomolgus monkeys. Blood samples were withdrawn at 0.083, 0.25, 0.5, 1, 2, 4, 8, and 24 h after intravenous administration and 0.25, 0.5, 1, 2, 4, 8, and 24 h after oral administration, using a needle syringe, from the cervical vein opened under isoflurane anesthesia. After centrifugation at 6,000$g$, 4°C for 10 min, the concentration of UD-051 was measured using LC-MS/MS (Shimadzu Corporation).

To assess the Nrf2 activity of UD-051 in mouse kidney tissue, UD-051 was orally administered to 6-wk-old female BALB/c mice at concentrations of 0.1, 0.3, 1, 3, or 10 mg/kg, and kidney tissues were collected at 1, 3, 6, 14, or 24 h after treatment. The mRNA expression of Nqo1 was measured using the method described below. Moreover, UD-051 was orally administered to 7-wk-old male C57BL/6J mice or 6-wk-old male SD rats at concentrations of 0.1, 0.3, 1, or 3 mg/kg for four consecutive days, and kidney tissues were collected 24 h after the final dose. Protein expression of NQO1 was measured using a simple Western blotting.

All animal experiments were approved by the Animal Care and Use Committee of Kumamoto University, Kumamoto, Japan (A2020-020, A2021-161, A2022-100), or the Animal Care and Use Committee of the Pharmaceutical Research Laboratory of UBE Corporation, Yamaguchi, Japan (P-18214, P-18232, P-18313, P-18229, P-19047, P-19268, P-21073, P-21145, P-21156). Animals were held in rooms with constant temperature and humidity and 12-h/12-h light cycles, and had free access to drinking water and standard chow.

### Real-time quantitative RT–PCR

Total RNA was isolated from mouse kidneys using NucleoSpin RNA (Takara) with homogenization. Reverse transcription and PCR amplification were performed using PrimeScript RT Reagent Kit with gDNA Eraser and SYBR Premix Ex Taq II (Takara), respectively, according to the manufacturer's recommended protocol. The sequences of primers used for qPCR are listed in Table S1.

### Urinary biomarkers

Mouse urine samples were collected for 24 h at the indicated time points using a metabolic cage (AS ONE Corporation). Urinary creatinine was measured by Jaffe's method (636-51011; Fujifilm) and used to normalize each urinary marker. Urinary protein was measured by the Bradford method (5000001; Bio-Rad). Urinary KIM-1 and clusterin concentrations were measured by Mouse Kidney Injury Magnetic Bead Panel 1 (MKI1MAG-94K). Urinary cystatin C and NGAL concentrations were measured using Mouse Kidney Injury Magnetic Bead Panel 2 (MKI2MAG-94K). Urinary TFF3 concentration was measured using Mouse TFF3 SimpleStep ELISA Kit (ab253228).

### Glomerular filtration rate

GFR was assessed as previously described (Kaseda et al, 2021).In brief, mice were anesthetized with isoflurane and injected with 7.5 mg/100$g$ body weight FITC-sinistrin (MediBeacon GmbH) through the subclavian vein. A transdermal GFR monitor (Medi-Beacon) was affixed directly to shaved skin on the dorsum of the animal, and levels of FITC-sinistrin were measured. Calculation of GFR was performed with MediBeacon software according to previously published methods.

### Plasma biomarkers

Mouse blood samples obtained from the inferior vena cava or tail were centrifuged at 800$g$, 4°C for 15 min, and blood plasma was collected. Plasma creatine, BUN, and indoxyl sulfate were measured by DRI-CHEM (Fujifilm), 7,180 biochemistry automatic analyzer (Hitachi), and PU-4180 HPLC Pump/FP-4020 Fluorescence detector (Jasco), respectively.

### Histological analysis

Kidney tissues were fixed in 10% formalin and embedded in paraffin. Tissue blocks were sliced into 2-$\mu$m thickness using a microtome and stained with PAS and Masson's trichrome. For PAS staining, after deparaffinization, sections were treated with 1% periodic acid for 15 min and were washed under running tap water. The sections were stained with Schiff's reagent for 15 min, followed by washing three times with sulfite water solution for 3 min, and washing with running tap water and distilled water. Samples were subsequently incubated with hematoxylin for 1 min and were dehydrated with ethanol and xylene. For Masson's trichrome staining, after deparaffinization, sections were treated with an equal mixture of 10% trichloroacetic acid solution and 10% potassium dichromate solution for 20 min and were washed under running tap water. Samples were subsequently incubated with Carazzi's hematoxylin solution for 1 min and were washed under running tap water followed by staining with 0.75% orange G solution for 1 min, Ponceau Xylidine–Acid Fuchsin for 20 min, 2.5% phosphotungstic acid solution for 30 min, and aniline blue for 15 min, washing twice with 1% acetic acid after each stain.

Tissues were imaged on a BZ-X700 microscope and analyzed by image analysis software (KEYENCE). Glomerulosclerosis score and fibrosis area were evaluated as reported previously (Kaseda et al, 2021; Omachi et al, 2021; Sannomiya et al, 2021). More than 100 random glomeruli per mouse were scored based on the following criteria, 0: no lesion; 1: expansion of mesangial area; 2: expansion of Bowman's epithelial cells, adhesion of glomeruli, and Bowman's capsule and partial sclerosis; 3: sclerotic area in 50–75% of glomeruli; and 4: sclerotic area in 75–100% of glomeruli. Fibrosis area was measured and normalized to tissue area at 10 different points (total area is 3,938,800 $\mu$m$^2$) in each mouse.

### Immunostaining

Paraffin-embedded mouse kidney sections (2 $\mu$m) were used for staining. After deparaffinization, sections were antigen-retrieved

for 20 min at 121°C with Dako Target Retrieval Solution (S2369; Agilent) or Proteinase K (S3020; Dako). Blocking was performed with Protein Block (X090930-2; Agilent), 3% BSA, or M.O.M. Blocking Reagent (FMK-2201; Vector Laboratories). The tissue sections were then incubated with the following primary antibodies overnight at 4°C: anti-albumin (ab19194; Abcam), biotinylated LTL (B-1325-2; Vector Laboratories), anti-megalin (ab76969; Abcam), anti-F4/80 (MCA497R; Bio-Rad), anti-α-SMA (ab5694; Abcam), anti-KIM-1 (AF1817; R&D Systems), anti-NQO1 (sc-393736; Santa Cruz Biotechnology), anti-cubilin (AF3700; R&D Systems), and anti-CD36 (A17339; ABclonal). After washing with PBS, HRP or fluorescent secondary antibodies were applied for 1 h at RT.

Tissues were imaged on a BZ-X700 microscope and analyzed by image analysis software (KEYENCE). To quantify the albumin localization in proximal tubules, at least 200 LTL-positive tubules from each mouse were examined. The strongly stained area of LTL (brush border) was defined as extracellular (intraluminal), and weakly stained area (cell bodies) as intracellular. The staining intensity of megalin, cubilin, and CD36 was measured and normalized to the positive area at 10 different points (total area is 3,938,800 $\mu m^2$) in each mouse. The F4/80- and α-SMA–positive area was measured and normalized to tissue area at 10 different points (total area is 3,938,800 $\mu m^2$) in each mouse. The number of KIM-1– and LTL-positive tubules was measured and normalized to the cortex area in each mouse.

### Glomerular RNA-seq

Glomeruli were isolated using magnetic beads. Briefly, mice were perfused with prewarmed 38 ml Hank's balanced salt solution (HBSS) and 2 ml HBSS with enzymatic digestion solution (300 U/ml Collagenase type II [Sigma-Aldrich], 1 mg/ml Proteinase E [Sigma-Aldrich], 50 U/ml DNase I [Invitrogen], and 8 × $10^7$ Dynabeads M-450 Tosylactivated [Invitrogen]). Kidneys were removed, minced into 1-$mm^3$ pieces, and digested in 2 ml enzymatic digestion buffer at 37°C for 20 min on a rotator. The digested kidneys were passed through a 200-$\mu$m cell strainer, and glomeruli were washed four times and collected using a magnetic particle concentrator (Invitrogen).

Total RNA from glomeruli was isolated and purified using RNeasy Plus Mini Kit (QIAGEN). The purity and integrity of the isolated RNA were checked by Epoch Microplate Spectrophotometer (BioTek) and Agilent 2100 BioAnalyzer. Poly(A)-selected cDNA libraries were generated using the TruSeq Stranded mRNA Library Prep kit (Illumina). The sequencing was performed using the NextSeq 500 system (Illumina) in 76-bp single-end reads. After adaptor trimming and quality check by Trim Galore (v0.5.0), sequencing reads were aligned to the mouse reference genome (mm10) using STAR (v2.6.0a). Gene expression profiles for each sample were measured as transcripts per million (TPM) using RSEM (v1.3.1). Differentially expressed genes, fold change of > 2 or < −2, *P* < 0.05, FDR < 0.05 (WT versus Alport vehicle, Alport vehicle versus Alport UD-051) were measured using DESeq2 (Etoh & Nakao, 2023) and subjected to heatmap analysis. TPM data were subjected to Gene Set Enrichment Analysis (v4.3.2). The RNA-seq data have been deposited in the DNA Data Bank of Japan (DDBJ) Sequence Read Archive under the accession number PRJDB11868. The RNA-seq data of WT and Alport

vehicle were described previously (Kaseda et al, 2021), but the animal experiment and analysis were conducted at the same time.

### Whole kidney RNA sequencing

Total RNA was isolated from mouse kidneys using Maxwell RSC simplyRNA Tissue Kit (Promega). The purity and integrity of the isolated RNA were checked by NanoDrop ONE and Bioanalyzer (Agilent). Poly(A)-selected cDNA libraries were generated using the NEBNext Poly(A) mRNA Magnetic Isolation Module and NEBNext Ultra II RNA Library Prep Kit for Illumina. The sequencing was performed using the NextSeq 500 system (Illumina) in 76-bp paired-end reads. Adaptor trimming, quality check, sequencing read alignment to the mouse reference genome (*Mus musculus* GRCm38 release-94), and gene expression profile analysis were performed using CLC Genomics Workbench (v20.0.4). Differentially expressed genes, fold change of > 2 or < −2, *P* < 0.05, FDR < 0.05 (WT versus Alport vehicle, Alport vehicle versus Alport UD-051) were measured using DESeq29 and subjected to heatmap analysis. TPM data were subjected to Gene Set Enrichment Analysis (v4.3.2). The RNA-seq data have been deposited in the DDBJ Sequence Read Archive under the accession number PRJDB18376.

### Transmission electron microscopy

Mouse kidneys were cut into 1-mm-thick sections and fixed in situ in 4% paraformaldehyde and 2.5% (wt/vol) glutaraldehyde in 0.1 M Hepes buffer (pH 7.4). Tissue was postfixed with 1% (wt/vol) osmium tetroxide, 1.5% (wt/vol) potassium ferrocyanide in 0.1 M cacodylate buffer for 1 h, followed by 1% (wt/vol) thiocarbohydrazide for a further hour. After washing, additional staining was performed in 1% (wt/vol) osmium tetroxide (1 h), followed by 1% (wt/vol) uranyl acetate overnight at 4°C. The final staining was performed at 60 °C with lead aspartate, pH 5.5, for 35 min. Samples were dehydrated in ethanol and infiltrated with TAAB 812 hard resin. Samples were sectioned (70–80 nm thickness) and examined using a Talos L120C TEM. More than 15 images of the glomerular capillaries were taken per mouse and used to measure the GBM thickness, FP width, and FP number per 1-$\mu$m GBM.

### Oxidized albumin ratio

Mouse blood samples obtained from the inferior vena cava or tail were mixed with 500 mM citrate buffer at a ratio of 9:1, and centrifuged at 800*g*, 4°C for 10 min to collect the plasma samples. After 5 $\mu$l of plasma was added to 495 $\mu$l of 50 mM phosphate buffer (pH 6.0), OAR was measured using electrospray ionization time-of-flight mass spectrometer (ESI-TOF MS) as previously described (Kato et al, 2021).

### Fluorescence polarization assay

UD-051 (3-(1,4-dimethyl-1H-benzo[d][1,2,3]triazol-5-yl)-3-(3-(((R)-2-ethyl-2,3-dihydro-[1,4]oxazepino[7,6-g]quinolin-4(5H)-yl)methyl)-4-methylphenyl)-2,2-dimethylpropanoic acid) (Diastereomer 1) was synthesized by the Pharmaceuticals Research Laboratory at UBE

Corporation. Seventy microliters of varying concentrations of UD-051 was added to 350 μl of buffer solution (20 mM Tris–HCl, 150 mM NaCl, 0.05% Tween-20, 5 mM DTT) containing 6 nM FITC-labeled NRF2 peptide (Invitrogen) and 0.2 mg/ml BSA. Then, 120 μl of solution was added to a 96-well plate and mixed with 80 μl of buffer solution containing 75 nM human FLAG-KEAP1 protein (ProCube system, Sysmex Corporation). After incubation at RT for 30 min, fluorescence polarization was measured at $\lambda$ex = 482 nm and $\lambda$em = 530 nm. The inhibition rate was calculated using the following equation:

$$\text{Inhibition rate} (\%) = 100 - \left[ \left( A_{sample} - A_{negative\ control} \right) / \left( A_{positive\ control} - A_{negative\ control} \right) \right] \times 100.$$

Wells without compound were used as a positive control, and wells without KEAP1 were used as a negative control.

### NQO1 activity assay

NQO1 activity was measured using a previously reported method with modifications (Prochaska & Santamaria, 1988; Fahey et al, 2004). Hepa1c1c7 cells (95090613; DS Pharma Biomedical) cultured in a 96-well plate with MEM$\alpha$ (135-15175; FUJIFILM) containing 10% FBS (10082; Gibco) and 1% Penicillin–Streptomycin–Amphotericin b (15240; Gibco) were treated with the indicated concentration of UD-051 for 48 h. After removing the medium and shaking with 50 μl of cell lysis buffer (9803; CST) containing protease inhibitor (11-873580001; Roche Diagnostics) for 20 min, reaction solution (25 mM hydrochloric acid, 0.07% albumin, 0.01% Tween-20, 2 U/ml glucose-6-phosphate dehydrogenase, 5 μM flavin adenine dinucleotide, 1 μM glucose-6-phosphate, 30 μM nicotinamide adenine dinucleotide phosphate, 0.03% 3-(4,5-dimethyl-2-thiazolyl)-2,5-diphenyltetrazolium bromide, 50 μM menadione) was added and leveled at RT for 5 min. They were followed by adding 50 μl of stop solution (0.3 mM dicoumarol, 5 mM potassium dihydrogen phosphate, pH 7.4), and absorbance was measured at 540 nm. The number of cells was measured using a CellTiter-Glo assay (G9242; Promega) using another plate seeded under the same conditions as above, and the absorbance was corrected.

### Small interfering RNA (siRNA) transfection

Hepa1c1c7 cells cultured in a 96-well plate with MEM$\alpha$ containing 10% FBS and 1% penicillin–streptomycin–amphotericin b were transfected with siRNA (Silencer Select Negative Control No.1 siRNA, 4390843, or Silencer Select Pre-Designed siRNA/S70522, 4390771; Thermo Fisher Scientific) using Lipofectamine (13778150; Thermo Fisher Scientific) diluted in Opti-MEM (31985070; Thermo Fisher Scientific). Twenty-four hours after transfection, cells were treated with UD-051 for 6 h at the indicated concentrations, then mRNA was collected, and the expression levels of Nrf2 and Nqo1 were measured. The sequences of primers used for qPCR are listed in Table S1.

### Statistics

All data are presented as the mean ± SE. The significance of the difference between the two groups was assessed using Student's unpaired two-tailed $t$ test. For comparisons of three or more groups, we used analysis of variance (ANOVA) with Dunnett's multiple comparisons test or Tukey's multiple comparisons test. Survival analysis was evaluated using a log-rank (Mantel–Cox) test. The correlation data were evaluated using the Pearson correlation coefficient. $P$-values < 0.05 were considered to be statistically significant.

## Data Availability

The RNA-seq data have been deposited in the DDBJ Sequence Read Archive under accession numbers PRJDB11868 and PRJDB18376.

## Supplementary Information

## Acknowledgements

We thank H Nishiyama and K Komori for the synthesis of UD-051, S Usuki, Y Mizukami, and K Watanabe for Next-Generation Sequencing, and S Forbes and A Mironov for TEM. This work was supported by the Japan Agency for Medical Research and Development (23ek0310017h0003 to H Kai), Japan Society for the Promotion of Science KAKENHI (JP23K14358 to S Kaseda, JP22H02810 to H Kai, and JP23K06165 to MA Suico), and Wellcome Trust Senior Fellowship (202860/Z/16/Z to R Lennon).

### Author Contributions

S Kaseda: conceptualization, data curation, formal analysis, funding acquisition, validation, investigation, visualization, methodology, and writing—original draft, review, and editing.
J Horizono: data curation, investigation, visualization, and methodology.
Y Sannomiya: data curation and investigation.
J Kuwazuru: data curation and investigation.
MA Suico: supervision, funding acquisition, validation, and writing—review and editing.
R Sato: data curation and investigation.
H Fukiya: data curation and investigation.
H Sunamoto: data curation and investigation.
S Ogi: data curation and investigation.
T Matsushita: data curation and investigation.
Y Koyama: data curation and investigation.
A Owaki: data curation and investigation.
H Tsuhako: data curation and investigation.
M Shiraga: data curation and investigation.
H Watanabe: conceptualization and methodology.
T Nakano: data curation and investigation.
B Davenport: data curation and investigation.
K Nozu: project administration and writing—review and editing.
M Yamamoto: resources.

T Shuto: resources, supervision, project administration, and writing—review and editing.

Y Tokunaga: resources, supervision, project administration, and writing—review and editing.

R Lennon: supervision, funding acquisition, project administration, and writing—review and editing.

K Onuma: conceptualization, data curation, formal analysis, validation, investigation, methodology, and writing—review and editing.

H Kai: conceptualization, supervision, funding acquisition, and writing—review and editing.

## Conflict of Interest Statement

S Ogi and K Onuma are co-inventors on patent WO2020/241853 A1 held by UBE Corporation. The other authors declare that they have no competing interests.

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
