## [Reviewer comments · Life Science Alliance]

Life Science Alliance

Efficacy of Nrf2 activation in a proteinuric Alport syndrome mouse model

Shota Kaseda, Jun Horizonono, Yuya Sannomiya, Jun Kuwazuru, Mary Suico, Ryoichi Sato, Hirohiko Fukiya, Hidetoshi Sunamoto, Sayaka Ogi, Takashi Matsushita, Yuimi Koyama, Aimi Owaki, Haruki Tshako, Masahiro Shiraga, Hiroshi Watanabe, Takehiro Nakano, Bernard Davenport, Kandai Nozu, Masayuki Yamamoto, Tsuyoshi Shuto, Yasunori Tokunaga, Rachel Lennon, Kazuhiro Onuma, and Hirofumi Kai

DOI: <https://doi.org/10.26508/lsa.202503330>

Corresponding author(s): Hirofumi Kai, Kumamoto University and Kazuhiro Onuma, UBE Corporation

Review Timeline:

Submission Date:	2025-03-29
Editorial Decision:	2025-05-15
Revision Received:	2025-05-21
Accepted:	2025-05-23

Scientific Editor: Tim Fessenden

Transaction Report:

May 15, 2025

RE: Life Science Alliance Manuscript #LSA-2025-03330-T

Prof. Hirofumi Kai
Kumamoto University
Molecular Medicine
5-1 Oe-honmachi
Kumamoto 862-0973
Japan

Dear Dr. Kai,

Thank you for submitting your manuscript entitled "Efficacy of Nrf2 activation in a proteinuric Alport syndrome mouse model". This manuscript was evaluated by three expert referees whose reports are appended to this email.

As you will see, all reviewers remark on the significance of these findings for the pathobiology and treatment of Alport syndrome. Reviewers 1 and 2 felt this work is suitable for publication in its present form and made minor suggestions which we invite you to consider. Reviewers 2 and 3 both noted that albumin levels were measured with nonspecific methods, and here we concur with Reviewer 2 that the observations support the central claims made from these measurements despite this lack of specificity. Reviewer 3 also noted that the timepoint analyzed in Fig 3I is not indicated. In view of the strong support from reviewers, we would be happy to publish your paper in Life Science Alliance pending this minor change and the final revisions necessary to meet our formatting guidelines.

To prepare your manuscript for eventual publication, please tend to the following:

- please upload your main manuscript text as an editable doc file.
- please upload your main and supplementary figures as single files; it is recommended to exclude figures from the manuscript text and upload them separately.
- please rename your supplementary figure from Appendix Figure S1 to Figure S1, etc.
- please add a Running Title and a Summary Blurb/Alternate Abstract in our system.
- please add ORCID ID for corresponding (and 2ndary and 3rd corresponding) author--you should have received instructions on how to do so.
- please add Keywords and a Category for your manuscript in our system.
- please add the X and Bluesky handles of your host institute/organization as well as your own or/and one of the authors in our system.
- please add authors' contributions to our system as well.
- please add your main and supplementary figure legends to the main manuscript text after the references section.
- please add the scale bar values in the legend for Figure S8F.
- please add callouts for Figures 3G, H; S4A-B; S5A-J; S6A-C; S8A-F; S9A-D and S10A-D to your main manuscript text.
- please remove the section called "The paper explained" on page 28.
- please incorporate supplementary methods in the main manuscript text.

A. FINAL FILES:

B. MANUSCRIPT ORGANIZATION AND FORMATTING:

Sincerely,

Reviewer #1 (Comments to the Authors (Required)):

Alport syndrome is a hereditary kidney disease often accompanied by hearing and eye defects. There are no targeted therapies for the disease, but some non-specific drugs that lower blood pressure and treat diabetes have shown efficacy for treating Alport syndrome. Here the authors show that a Nrf2 activator that they discovered, the Keap1-Nrf2 protein-protein interaction inhibitor UD-051, is effective at slowing kidney disease progression in a C57BL/6J X-Linked Alport syndrome mouse model. In combination treatment with the Angiotensin II receptor blocker losartan, there was an apparent synergistic effect at slowing the decline of kidney function and increasing survival time by 66%. This finding will be of great interest to those in the kidney disease field, despite the drawback that a clinical trial with bardoxolone methyl, a drug that also activates Nrf2 but by a different mechanism, was halted due to safety concerns. Time will tell whether the clinical trials community will embrace the testing of a different drug that increases Nrf2 activation.

Perhaps due to the bardoxolone methyl trial issues, the authors generated an unusually enormous amount of data to support their studies of the mouse model. The data are truly comprehensive and touch on diverse aspects of disease progression, including effects on urinary total protein and albumin, blood creatinine and urea nitrogen, impact on histopathology by light and electron microscopy, transcriptomes, inflammation, biomarkers for kidney tubular injury, and many others. Multiple gene edited mouse models were used to solidify UD-051's mechanism of action via Nrf2 activation. There are no missing critical experiments, and the complex collection of data are high quality and presented in very accessible formats.

I have a few suggestions to improve the paper.

1. There are some abbreviations that are not spelled out. CDDO-Im in the abstract should be explained, as should OAR on its first use (page 13).
2. Towards the end of the abstract, do the authors mean "insight into the efficacy of Nrf2 activation in Alport syndrome"?
3. Fig. S6 shows CD36 expression, but in the text the authors refer to it as showing FcRn expression.

Reviewer #2 (Comments to the Authors (Required)):

This is a large and detailed look at a topical question that has been taxing researchers. Human disease trials with bardoxolone created controversy over whether it and related drugs (the actions of which include Nrf2 activation) can be effective in renal disease, or whether apparent effects on renal function are illusory; and how their poorly understood effects might be explained. The authors describe the effects of new Nrf2-activating compound UD-051, which inhibits the interaction of Nrf2 with its ubiquitinating enzyme Keap1, on a mouse model of the human genetic disease Alport Syndrome.

A large number of well-planned and conducted experiments are very clearly presented. There are two striking and important findings.

(1) A striking beneficial effect on not only measured GFR but also on survival is shown, particularly when used in combination with current best therapies with ACE inhibition or ARB (here the ARB losartan).

(2) Evidence that a large part, if not all, of the benefits from this drug are achieved via protection of the tubulointerstitium, rather than the glomerulus. This is an important and striking observation. Previous therapies have purported to do this, but have been unsuccessful, or have been shown to have possibly more important direct effects on the glomerulus.

It would be useful to confirm that the late deaths in the combined-treatment experiments (Fig 6) were from renal failure, either from terminal blood measurements or kidney histology, if this data is available.

Other comments

Alport models have been the subject of many studies, and the strain chosen here is accepted to be a good model of human X-linked Alport syndrome. It and other Alport models have also been used to test non-specific therapies for proteinuric renal diseases that may be applicable much more widely.

The authors have not excluded direct effects of their or other Nrf2 activation strategies on the glomerulus, but confirm that treatment is associated with lowering of megalin expression in PCT cells, and with reduced intracellular albumin identification. The lack of effect on cubulin is an interesting contrast. Biomarker studies tend to confirm tubular effects, including reductions in tubular injury.

Urinary albumin has not been measured by specific assays, only total protein, but the dose relationships, controls and biomarker observations do support a major effect of UD-051 on tubular reabsorption of protein. Therefore proteinuria observed with Nrf2 activators, or at least with UD-051, may not have the same adverse implications as glomerular proteinuria; these drugs may reduce adverse consequences of glomerular proteinuria.

The observations here shed no light on the mechanism of GFR increase seen (or in some instances only presumed from Creatinine changes) with Nrf2 activators.

The results of these observations and of others with Nrf2 activators suggest that combination with SGLT2 inhibition would (also) be beneficial. Nrf2 activation is reported to upregulate SGLT2 expression, and increase blood pressure and fluid (salt) retention. SGLT2 inhibition appears to have wide benefits in proteinuric renal disease. Murine studies suggest the power of adding it to other agents in an Alport model. Human studies show that it reduces proteinuria, and reduces disease progression in diverse types of CKD, whether or not there is proteinuria. This will be worthy of further study.

Side effects have been a significant concern in human bardoxolone trials. Animals in active treatment arms with different Nrf2 activation methods experienced reduced weight gain, as has been seen in human studies. It is therefore reassuring that there was a striking, dose-related increase in survival in the dual-therapy study in this paper.

Inconsistency of effects of different approaches or doses of agents that activate Nrf2 is a problem for both understanding, and confidence about being able to achieve a clinically useful therapy. The authors present arguments in favour of this being a question of strength or duration of Nrf2 activation by different agents, but this is mostly supposition. Other-target effects and other mechanisms must also be possible.

Gene expression studies provide an interesting companion to the major observations in the paper, generating hypotheses about mechanism and locus of action.

Reviewer #3 (Comments to the Authors (Required)):

In this original manuscript the authors test the use of a KEAP1/NRF2 protein-protein interaction inhibitor for the treatment of Alport syndrome. NRF2 is a cytoprotective detoxifying and antioxidant pathway that has been of interest for treatment of kidney diseases. Unfortunately, clinical trials in diabetic kidney disease and Alport syndrome have not shown efficacy of bardoxolone methyl (an NRF2 inducer) and in fact may have revealed harmful effects such as increase in proteinuria that have been replicated in some animal studies, although considerable controversy exists. Since bardoxolone binds to KEAP1 irreversibly, noncovalent reversible PPI inhibitors may have a different effect.

The authors show that Keap1 knockdown mice bred with Alport mice does not improve overall injury or survival, although a slight increase in proteinuria was seen. Meanwhile, CDDO-Im, a bardoxolone analog, slightly reduced some inflammatory cytokines but didn't affect overall disease progression.

The authors then show that a new PPI inhibitor, UD-051, ameliorated progression of Alport syndrome disease in mice with genetic loss of Col4a. This drug is orally bioavailable and upregulates NRF2 activity in mice and other animals. GFR, glomerular injury, and fibrosis is reduced with long-term administration of UD-051. Transcriptome analysis shows major changes in tubular genes but less so in glomerular genes, suggesting a primary effect on tubules. As in clinical trials, proteinuria was transiently increased by UD-051. This was shown to potentially be due to downregulation of tubular albumin uptake. In combination with losartan, UD-051 improved survival and losartan prevented the increase in proteinuria. UD-051 effects are dependent on NRF2 as knockout mice lack a response. Oxidized albumin could be a biomarker for NRF2 activity, which may be important because there are dose-dependent (hormetic) effects of NRF2 induction that may need careful titration.

Generally this is important work and contributes to our understanding of Nrf2 in CKD. Some of the mechanistic insights are less well-supported, and because it shows an opposite result to other published studies, controversy and skepticism will remain regarding Nrf2 in glomerular diseases. However, the general phenotypes shown in his work are impressive, and the idea that in this study the tubular compartment rather than the glomerular compartment is being protected, and the idea of using oxidized albumin as a biomarker for Nrf2 are all interesting. Several comments should be addressed:

Major comments:

-- The proteinuria and albuminuria are determined by Bradford and Coomassie brilliant blue assays respectively. These may be less accurate, and I'm not aware that CBB is specific for albumin. Would suggest using an albumin ELISA (with results normalized to urine creatinine). The authors should consider validating a few of their key albuminuria findings with ELISA (Suggest Fig 1 wk 12 and 22 (just the alport mice); Fig 3, wk 12 and 22; and Fig 6, wk 14 and 22). If this cannot be done, would suggest changing all albuminuria labels to proteinuria instead. Finally why do the authors think losartan is not decreasing proteinuria at 14w but does so at 22w?

-- What are the proteinuria results for Supp fig 12?

-- Figure 3I does not tell the timepoint being analyzed. If at 22 weeks, this may be the wrong timepoint to assess since proteinuria was not different at 22 weeks but rather the increases were seen around 12 weeks.

-- If UD-051 is reducing megalin, then b2-microglobulin urinary excretion should be higher. While I acknowledge that cystatin c is one marker for tubular reuptake, this finding of lower b2-microglobulin seems to argue against the megalin hypothesis - what is the explanation for this?

--the seq data is really a large amount for the reader to digest. As the authors indicate, some pathways are inversely correlated in the ud051 and alport only group (page 11 - "many of the GO altered by UD-051...inversely correlated with the GO altered in Alport vehicle). The authors should specifically name the pathways that are being inversely correlated.

--the authors discuss that the tubular compartment is the main target of UD051. This doesn't seem entirely true as the glomerular compartment did show mild improvements by glomerular scoring and some of the seq data shows this as well, albeit to a lower extent than the tubules. However, the general concept of tubular protection is intriguing, because a large amount of published data shows that Nrf2 protects against tubulointerstitial diseases like AKI. (It is direct glomerular injury that in many cases are made worse by Nrf2 activity.) The authors should consider expanding the discussion of this distinction, which I believe could be an important distinction here.

Minor comments:

-- would be good to show mRNA for nqo1 in CDDO-Im experiments as a comparison to UD effects.

-- in glomerular scoring it is strange to see that % of glomeruli with the score of 3 is numerically less than score of 2 or 4. Is this because when sclerosis is seen it is always severe?

-- in the proteinuria discussion, the authors should point out that clinical trials of bardoxolone universally show increased

proteinuria in recipients (CARDINAL, BEAM, BEACON). this is, and will remain, an overall concern with Nrf2.

-- what is the rationale for decreasing losartan doses during the experiment, from 250 down to 125?

-- a legend or key is required to define groups in 6G-M

Response to Reviewers' Comments

Reviewer #1

I have a few suggestions to improve the paper.

1. There are some abbreviations that are not spelled out. CDDO-Im in the abstract should be explained, as should OAR on its first use (page 13).

Answer:

The CDDO-Im (pp. 4 & 6) and OAR (p. 13) abbreviations have been spelled out.

2. Towards the end of the abstract, do the authors mean "insight into the efficacy of Nrf2 activation in Alport syndrome"?

Answer:

We added the word "activation" in the Abstract.

3. Fig. S6 shows CD36 expression, but in the text the authors refer to it as showing FcRn expression.

Answer:

We changed the "FcRn" to "CD36" (p. 10).

We thank this Reviewer for the encouraging comments and corrections to the manuscript.

Reviewer #2

1. It would be useful to confirm that the late deaths in the combined-treatment experiments (Fig 6) were from renal failure, either from terminal blood measurements or kidney histology, if this data is available.

Answer:

Since blood and kidney samples were not collected for the survival study, we are unable to investigate the cause of death. However, all of the animals that died or were humanely sacrificed in the survival experiment showed severe kidney atrophy and blackening, indicating kidney dysfunction, and strongly suggesting that the cause of death was renal failure.

2. The results of these observations and others with Nrf2 activators suggest that combination with SGLT2 inhibition would also be beneficial. Nrf2 activation is reported to upregulate SGLT2 expression, and increase blood pressure and fluid (salt) retention. SGLT2 inhibition appears to have wide benefits in proteinuric renal disease. Murine studies suggest the power of adding it to other agents in an Alport model. Human studies show that it reduces proteinuria and reduces disease progression in diverse types of CKD, whether or not there is proteinuria. This will be worthy of further study.

Answer:

We have investigated the effect of the combination of the SGLT2 inhibitor dapagliflozin and the RAS inhibitor Olmesartan or losartan in a separate study. The paper is currently under review. We found that dapagliflozin has an add-on effect to losartan but not to Olmesartan, which is a strong anti-proteinuric drug. We agree that it will be insightful to study the combination of UD-051+Olmesartan or UD-051+Dapagliflozin+losartan.

3. Inconsistency of effects of different approaches or doses of agents that activate Nrf2 is a problem for both understanding and confidence about being able to achieve a clinically useful therapy. The authors present arguments in favour of this being a question of the strength or duration of Nrf2 activation by different agents, but this is mostly supposition. Other-target effects and other mechanisms must also be possible.

Answer:

We added the statement “Multiple-target effects and mechanisms may be possible. Based on our observations, we hypothesize that...” in the Discussion on page 17, line 4.

We thank this Reviewer for the helpful comments and suggestions.

Reviewer #3

1. The proteinuria and albuminuria are determined by Bradford and Coomassie brilliant blue assays, respectively. These may be less accurate, and I'm not aware that CBB is specific for albumin. Would suggest using an albumin ELISA (with results normalized to urine creatinine). The authors should consider validating a few of their key albuminuria findings with ELISA (Suggest Fig 1, wk 12 and 22 (just the Alport mice); Fig 3, wk 12 and 22; and Fig 6, wk 14 and 22). If this cannot be done, would suggest changing all albuminuria labels to proteinuria instead. Finally, why do the authors think losartan is not decreasing proteinuria at 14w but does so at 22w?

Answer:

We separated the urinary proteins by molecular weight in an SDS-PAGE gel stained with CBB. We quantified only the ~66 kDa band for albumin. Although the major protein in Alport mouse urine is albumin, as the reviewer pointed out, the presence of other proteins cannot be completely ruled out. Therefore, we removed the albuminuria data (Fig 3H) and showed only the proteinuria data.

Regarding the effect of losartan on proteinuria (not decreasing proteinuria at 14w but does so at 22w), we believe that this is because the progression of renal failure in Alport mice (B6.Col4a5-G5X mice) accelerates slowly in the early stages, and rapidly in the later stages, making it easier to observe the therapeutic effects in the later stages.

2. What are the proteinuria results for Supp fig 12?

Answer:

We added this result in Fig S12E and the following description on page 13, lines 9-10. "Losartan significantly suppressed proteinuria at 18 weeks of age, but did not prevent the subsequent increase in later stages. In contrast, combination therapy significantly reduced the increase in proteinuria at 24 weeks."

3. Figure 3I does not tell the time point being analyzed. If at 22 weeks, this may be the wrong time point to assess since proteinuria was not different at 22 weeks, but rather the increases were seen around 12 weeks.

Answer:

Because we collected the kidneys from UD-051-treated Alport mice at 22 weeks, we were not able to perform tissue analysis on 12-week-old mice, and there are limitations regarding the mechanism of increased proteinuria. The final sentence of the discussion on proteinuria was

revised to read as follows: (p. 16, last sentence). “Although further investigation is needed to clarify the mechanism, based on these reports and the current data, we hypothesize that the increase in proteinuria induced by UD-051 is non-pathologic, Nrf2-dependent, and independent of the progression of kidney disease.”

4. If UD-051 is reducing megalin, then b2-microglobulin urinary excretion should be higher. While I acknowledge that cystatin C is one marker for tubular reuptake, this finding of lower b2-microglobulin seems to argue against the megalin hypothesis - what is the explanation for this?

Answer:

Because we measured urinary b2-microglobulin at 22 weeks, we should also take into consideration the effect of UD-051 on improving renal pathology, not only its effect on megalin. As shown in Figure 2, UD-051 significantly suppressed the decline in renal function and progression of renal pathology in Alport mice, so the level of b2-microglobulin should also be suppressed in the UD-051-treated group, similar to other urinary markers. However, our results showed almost no difference between the vehicle and the treated group, suggesting that UD-051 increased the urinary b2-microglobulin level. Furthermore, previous reports have shown that megalin-mediated reabsorption is less effective for cystatin C than for b2-microglobulin (p. 16). Therefore, the decrease in cystatin C due to the improvement of renal pathology exceeded the increase in cystatin C due to the suppression of megalin.

5. The seq data is really a large amount for the reader to digest. As the authors indicate, some pathways are inversely correlated in the UD-051 and Alport only group (page 11 - "many of the GO altered by UD-051...inversely correlated with the GO altered in Alport vehicle). The authors should specifically name the pathways that are being inversely correlated.

Answer:

We modified the sentence to “In contrast, many of the GO altered by UD-051 in the whole kidney were inversely correlated with the GO (Dagger) altered in Alport-vehicle, such as the metabolic processes and immune response pathways” on page 11.

6. The authors discuss that the tubular compartment is the main target of UD051. This doesn't seem entirely true as the glomerular compartment did show mild improvements by glomerular scoring and some of the seq data shows this as well, albeit to a lower extent than the tubules. However, the general concept of tubular protection is intriguing because a large amount of published data shows that Nrf2 protects against tubulointerstitial diseases like AKI. (It is direct glomerular injury that in many cases, is made worse by Nrf2 activity.) The authors should

consider expanding the discussion of this distinction, which I believe could be an important distinction here.

Answer:

Alport mice develop spontaneous proteinuria and glomerulosclerosis, followed by inflammation, fibrosis, and other renal tubular damage, leading to end-stage renal failure. UD-051 dose-dependently attenuated the glomerulosclerosis in Alport mice, reducing the number of glomeruli showing severe sclerosis (Score 4) and increasing the proportion of glomeruli showing moderate sclerosis (Score 2), but did not significantly increase the number of glomeruli showing mild sclerosis (Score 1) (Fig 2G). We added these statements on p15, “In contrast to a partial protective effect in the glomerulus, UD-051 significantly ameliorated tubular injury, inflammation, fibrosis, and abnormalities of gene expression in whole kidney tissue. Although UD-051 was inferior to losartan in suppressing the decline in GFR and progression of glomerulosclerosis, it inhibited the onset of subsequent fibrosis and end-stage renal failure more than losartan (Fig 6B, G-J). These results suggest that the main therapeutic target for UD-051 is the tubular injury common to most CKDs, rather than the glomerular filtration defects characteristic of some glomerular diseases, including Alport syndrome”.

Minor comments:

1. It would be good to show mRNA for Nqo1 in CDDO-Im experiments as a comparison to UD effects.

Answer:

We have shown this data in our previously published paper (Kaseda S et al., Kidney360. 2021), so we referenced it in this study.

2. In glomerular scoring, it is strange to see that % of glomeruli with the score of 3 is numerically less than a score of 2 or 4. Is this because when sclerosis is seen, it is always severe?

Answer:

Because glomerular sclerosis progresses irreversibly, we think that after the glomerular capillary wall and Bowman's capsule epithelium have adhered to each other and glomerular sclerosis has formed, the progression is rapid, and the percentage of glomeruli showing the intermediate stage (score 3) is low.

3. In the proteinuria discussion, the authors should point out that clinical trials of bardoxolone

universally show increased proteinuria in recipients (CARDINAL, BEAM, BEACON). This is, and will remain, an overall concern with Nrf2.

Answer:

We included the following sentence in the Discussion on page 16. “Clinical trials of bardoxolone universally showed a transient increase of proteinuria in recipients, including Alport syndrome, which remains a concern with Nrf2 activation (Warady et al, 2022).”

4. What is the rationale for decreasing losartan doses during the experiment, from 250 down to 125?

Answer:

We added the following sentences in the Materials and Methods on page 19.

“Alport mice exhibit increased water intake as the disease progresses. Therefore, we reduced the concentration of losartan accordingly (Yokota et al, 2018; Kaseda et al, 2021; Omachi et al, 2021; Sannomiya et al, 2021)”.

5. A legend or key is required to define groups in 6G-M

Answer:

We added the following legend to Fig. 6G-M on p. 42.

“(G-M) Colors represent WT (black), Alport-vehicle (gray), and Alport-UD-051 (green)”.

We thank Reviewer 3 for the insightful comments and suggestions.

May 23, 2025

RE: Life Science Alliance Manuscript #LSA-2025-03330-TR

Dr. Hirofumi Kai
Kumamoto University
Department of Molecular Medicine
5-1 Oe-honmachi
Kumamoto 862-0973
Japan

Dear Dr. Kai,

Thank you for submitting your Research Article entitled "Efficacy of Nrf2 activation in a proteinuric Alport syndrome mouse model". It is a pleasure to let you know that your manuscript is now accepted for publication in Life Science Alliance. Congratulations on this interesting work.

DISTRIBUTION OF MATERIALS:

Again, congratulations on a very nice paper. I hope you found the review process to be constructive and are pleased with how the manuscript was handled editorially. We look forward to future exciting submissions from your lab.

Sincerely,
